# LeSTD: LLM Compression via Learning-based Sparse Tensor Decomposition

**Yi Li[1], Zhichun Guo[2], Miao Yin[3], Bingzhe Li[1]**

[1] University of Texas at Dallas, {`Yi.Li3, Bingzhe.Li`}@utdallas.edu
[2] Independent Researcher, `zcguo.work@gmail.com`
[3] University of Texas at Arlington, `miao.yin@uta.edu`

## Abstract

Large language models (LLMs) deliver the impressive capability, while their parameter scales hinder the deployment ability. Post-training matrix/tensor decomposition offers a promising strategy to alleviate this by exploiting structural redundancies within model weights. However, it faces the critical dense core bottleneck. This bottleneck caps achievable compression level as dense core tensor becomes a new storage burden. To solve this, we introduce LeSTD (**Le**arning-based **S**parse **T**ensor **D**ecomposition), a two-stage, data-free compression framework. LeSTD first learns a high-quality shared basis for model weights, then applies a theoretically-ground pruning mechanism: guided by a derived closed-form importance score, to create an ultra-sparse core tensor. Therefore, resulting a superior compression-accuracy trade-off: LeSTD achieves substantially higher compression ratios than dense-core methods without sacrificing performance. Experiments on the LLMs up to 30B parameters confirm that LeSTD consistently attains lower perplexity and higher task accuracy at matched compression levels, and critically, maintains strong performance under aggressive compression where prior methods degrade. Operationally, LeSTD executes the inference directly in its compressed domain, delivering significant throughput gains on standard hardware without requiring any custom kernels.

## 1 Introduction

Transformer-based Large Language Models (LLMs) (Vaswani et al., 2017) form the backbone of contemporary AI, demonstrating remarkable performance in natural language understanding (Nie et al., 2019; Gatt & Krahmer, 2018), code generation (Gu, 2023), and mathematical reasoning (Zeng et al., 2023; Imani et al., 2023). However, these advances are accompanied by an ever-growing model scale, with models comprising hundreds of billions of parameters. This imposes formidable storage and memory demands (Kwon et al., 2023; Hu et al., 2024), limiting their deployment in resource-constrained environments and necessitating effective model compression strategies.

Among various paradigms, post-training LLM compression approach have emerged as practical solutions as it can avoid the expensive retraining. This paradigm includes the pruning (Ma et al., 2023; Fu et al., 2024), quantization Liu et al. (2023; 2024); Zhao et al. (2024), and low-rank decomposition (Wang et al., 2024b; Yuan et al., 2023). Among all these different approaches, even the simplest low-rank approach, e.g., Singular Value Decomposition (SVD), has shown the promise on individual weight matrices (Wang et al., 2024b), demonstrates strong application potential.

Most existing low-rank methods (Yuan et al., 2023; Wang et al., 2024b) adopt a matrix-by-matrix compression strategy. While this approach is intuitive, it overlooks higher-order correlations and the structural redundancies spanning related weight matrices (Szekely et al., 2024). Consequently, it misses the opportunity to treat them as a unified object, which could further improve the achievable compression ratio. Moreover, vast studies have shown that multiple attention heads (MHA) within a single Transformer layer often learn the redundant patterns (Michel et al., 2019; Voita et al., 2019; Yang et al., 2024), highlighting additional opportunities for joint compression.
**Motivation.** Achieving joint compression is non-trivial. A fundamental question is whether a shared structure truly exists across heads within one layer? Intuitively, because all heads in a layer process

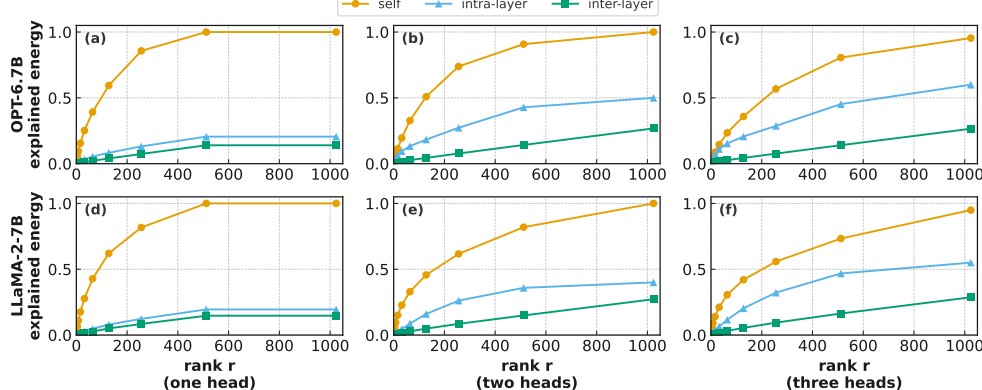

Figure 1: Subspace-sharing diagnostic on OPT-6.7B and Llama2-7B across multiple heads. For each column, we compute a rank-$r$ orthonormal basis from the concatenated weights $[\boldsymbol{W}^Q|\boldsymbol{W}^K|\boldsymbol{W}^V|(\boldsymbol{W}^O)^\top]$ of layer-15 using: (Figure (a),(d)) one head (head 0), (Figure (b), (e)) two heads (heads 0, 1), and (Figure (c), (f)) three heads (heads 0, 1, and 2), and evaluate the explained energy on self, intra-layer (other heads in layer 15), and inter-layer (layer-25 head 0), with $r \in \{2, 4, 8, 16, 32, 64, 128, 256, 512, 1024\}$. The top row shows OPT-6.7B and the bottom row shows Llama2-7B.

the same input distribution, their weight matrices might lie in (or near) a common subspace.

**Diagnostic.** To test this hypothesis, we design a diagnostic that quantifies subspace sharing. Core idea is to learn a basis from one attention head and assess how well it transfers to other heads. Concretely, for a given head $i$ in layer $l$, we construct its complete linear transformation by concatenating its constituent Query ($\boldsymbol{W}^Q$), Key ($\boldsymbol{W}^K$), Value ($\boldsymbol{W}^V$) and Output ($\boldsymbol{W}^O$) weights into a single matrix: $\boldsymbol{H}_{l,i} = [\boldsymbol{W}^Q|\boldsymbol{W}^K|\boldsymbol{W}^V|(\boldsymbol{W}^O)^\top] \in \mathbb{R}^{d_{\text{model}} \times 4 \cdot d_{\text{head}}}$. We then perform an SVD and extract top-$r$ left singular vectors to form an orthonormal basis $\boldsymbol{U}^{(r)}$. This basis is used to project and reconstruct three targets: (i) the source matrix itself (**self**), (ii) another head $j$ ($j \neq i$)'s matrix from the same layer (**intra**), and (iii) a head's matrix from a different layer $k$ ($k \neq l$) (**inter**). Reconstruction quality is measured by the *explained energy* (Greenacre et al., 2022; Wiskott, 2013), defined as $1 - E^2$, where $E = \frac{\|\boldsymbol{H} - \hat{\boldsymbol{H}}\|_F}{\|\boldsymbol{H}\|_F}$ is relative Frobenius reconstruction error (Tong et al., 2021) and $\hat{\boldsymbol{H}} = \boldsymbol{U}^{(r)}(\boldsymbol{U}^{(r)})^\top \boldsymbol{H}$ is reconstructed matrix (Eckart & Young, 1936). Explained energy quantifies how much of a target matrix's total information is retained when projected onto another head's basis, a higher value indicates stronger overlap between two subspaces and thus greater subspace sharing. The curves in Figure 1 directly address the central question posed in our motivation: does a shared subspace exist across heads within the same layer? The answer is affirmative. As the subspace rank $r$ and the number of heads increase, explained energy for **intra** systematically rises and remain consistently above **inter**. This pattern shows that a basis learned from one or more heads capture nontrivial portions of the variation of other heads in the same layer, and that captured portion grows as more dimensions are allowed. Hence, there is real, layer-local structure that different heads share. At the same time, for a single head, the intra-layer curves saturate well below the self ceiling. However, by examining layers with a larger number of heads, we find that the shared similarity between heads in terms of explained energy increases as the number of heads grows. Geometrically, this suggests that the corresponding subspaces exhibit greater overlap when more heads are present. This trend highlights additional untapped structure and indicates the potential for new methods that more effectively exploit the unexplained energy.

These two facts together validate our motivation and revealed a research gap: shared structure does exist, but exploiting it requires a mechanism that is richer than matrix-by-matrix (single-head) transfer. This mandates a joint optimization approach to identify a common subspace that can efficiently represent all heads simultaneously. Recent methods, such as the TensorLLM (Gu et al., 2025), apply a global Tucker decomposition to the MHA block and successfully identify a high-quality shared basis. While this represents a clear step forward beyond matrix-by-matrix compression, it also introduces a critical and unresolved limitation:

**Dense Core Bottleneck.** Tucker decomposition factorizes a tensor into a set of compact factor matrices and a core tensor. In our setting, the factor matrices capture low-dimensional shared subspaces

along each mode, while the core tensor encodes the remaining cross-mode interactions. While the factor matrices can be relative small, the core tensor remains fully dense (Ahmadi-Asl et al., 2021; Wang & Yang, 2022). To preserve model accuracy, the Tucker ranks must be sufficiently large, but the size of the core tensor grows polynomial with these ranks. This dense core rapidly becomes the new storage bottleneck, imposing a hard limit on the achievable compression ratio.

This limitation exposes a fundamental gap: current tensor-based methods reduce dimensionality but fail to eliminate redundancy within the compressed latent space itself. To achieve the truly high-ratio compression, we must moving beyond low-rank approximation and into the domain of sparse tensor representation (Park et al., 2021).

To fill this critical gap, we propose LeSTD (**Le**arning-based **S**parse **T**ensor **D**ecomposition), a framework that synergistically combines iterative basis optimization with the learned core tensor sparsity. LeSTD operates in two stages: first, it optimizes a high-quality shared basis for all attention heads; second, it learns an ultra-sparse representation for the core tensor within that basis. This integrated approach yields a representation that is both more accurate and vastly more compact. Main contributions of LeSTD are as follows:

1. The proposal of LeSTD, a data-free post-training compression framework where Stage I learns a high-quality shared basis via iterative optimization, and Stage II introduces a principled pruning strategy to create an ultra-sparse core tensor.
2. We provide a theoretical justification for the magnitude-based pruning in the Tucker-decomposed latent space. We derive a closed-form importance score for each core element, directly linking its magnitude to its impact on the Frobenius reconstruction error. This allows for a principled, rather than purely heuristic, sparsification of the core
3. We demonstrate how inference can operate directly on the compressed representation, avoiding the full weight reconstruction. It reduces arithmetic complexity and delivers practical throughput gains (*tokens/sec*) measured directly within the standard Transformers library (Wolf et al., 2020), requiring no specialized hardware or custom kernels.
4. Across GPT-J (6B), Llama2 (13B), and OPT (30B) on WikiText-2, MathQA, GSM8K, and TruthfulQA, LeSTD consistently outperforms baselines at matched size fractions: maintaining higher accuracy under strong compression and delivering competitive-to-superior throughput.

## 2   BACKGROUND AND PRELIMINARY

**Notation.** We denote tensors by uppercase calligraphic letters, e.g., $\boldsymbol{\mathcal{X}} \in \mathbb{R}^{I_1 \times I_2 \times \cdots \times I_N}$, matrices by uppercase bold letters, e.g., $\boldsymbol{X} \in \mathbb{R}^{I_1 \times I_2}$, and vectors by the lowercase bold letters, e.g., $\boldsymbol{x} \in \mathbb{R}^{I_1}$. The Frobenius norm of a tensor (Defant & Floret, 1992) is $\|\boldsymbol{\mathcal{X}}\|_F = \sqrt{\sum_{i_1,\ldots,i_N} x_{i_1,\ldots,i_N}^2}$. The mode-$n$ unfolding of $\boldsymbol{\mathcal{X}}$ is a matrix $\boldsymbol{X}_{(n)} \in \mathbb{R}^{I_n \times (\prod_{k \neq n} I_k)}$ whose columns are the mode-$n$ fibers of the tensor.

**Transformer Architecture.** Modern LLMs typically employ a decoder-only Transformer architecture (Vaswani et al., 2017; Fu et al., 2023), composed of a stack of identical layers, as shown at the right of Figure 2. Each layer contains a Multi-Head Attention (MHA) block and a Feed-Forward Network (FFN). MHA block allows the model to jointly attend to information from different representation subspaces. For the $i$-th attention head, the query

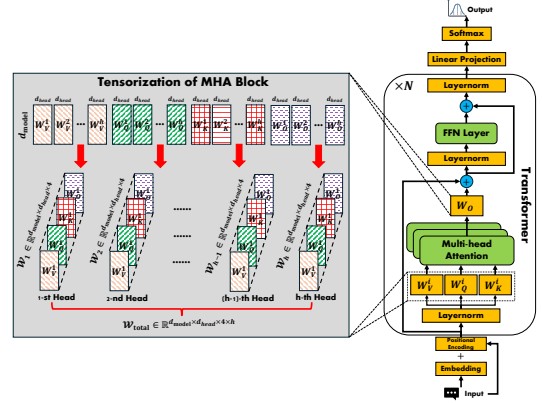

Figure 2: An illustration of the Transformer architecture (right) alongside the corresponding tensorization process used in our scheme (left).

$\boldsymbol{Q}$, key $\boldsymbol{K}$, and value $\boldsymbol{V}$ inputs are linearly projected via weighted matrices $\boldsymbol{W}_i^Q, \boldsymbol{W}_i^K, \boldsymbol{W}_i^V \in \mathbb{R}^{d_{\text{model}} \times d_{\text{head}}}$, $1 \leq i \leq h$, and $d_{\text{head}} = \frac{d_{\text{model}}}{h}$ is the per-head dimension:

$$\text{head}_i = \text{Att}(\boldsymbol{Q}\boldsymbol{W}_i^Q, \boldsymbol{K}\boldsymbol{W}_i^K, \boldsymbol{V}\boldsymbol{W}_i^V) = \text{softmax}\left(\frac{(\boldsymbol{Q}\boldsymbol{W}_i^Q)(\boldsymbol{K}\boldsymbol{W}_i^K)^\top}{\sqrt{d_{\text{head}}}}\right)(\boldsymbol{V}\boldsymbol{W}_i^V) \quad (1)$$

Figure 3: Overview of LeSTD. Left: MHA weights are tensorized. Middle: An iterative optimization finds high-quality shared orthogonal factor matrices ($U^{(n)}$) and a dense core ($\mathcal{G}_i$) for each head. Right: A pruning mechanism enforces extreme sparsity on the core tensors.

To improve stability and training efficiency, residual connections (Xie et al., 2023) and layer normalization (Xiong et al., 2020) are usually applied, though omitted here for clarity. The outputs of $h$ attention heads are concatenated and projected back into the latent space via an output matrix $W_O$:

$$\text{MultiHead}(Q, K, V) = \text{Concat}(\text{head}_1, ..., \text{head}_h)W_O \tag{2}$$

Thus, MHA block involves four weight matrices: the query projection $W^Q$, key projection $W^K$, value projection $W^V$, and the output projection $W^O$. We refer to these as the MHA weight matrices, which constitute the target of our compression.

**Tensor Operations and Decompositions.** The $n$-mode product of a tensor $\mathcal{X} \in \mathbb{R}^{I_1 \times I_2 \times \cdots \times I_N}$ with a matrix $U \in \mathbb{R}^{J_n \times I_n}$ is a tensor $\mathcal{Y} \in \mathbb{R}^{I_1 \times \cdots \times J_n \times \cdots \times I_N}$, denoted as $\mathcal{Y} = \mathcal{X} \times_n U$. Element-wise, this is defined as $(\mathcal{Y})_{i_1,...,j_n,...,i_N} = \sum_{i_n=1}^{I_n} x_{i_1,...,i_n,...,i_N} u_{j_n,i_n}$. This operation can be understood as multiplying the matrix $U$ with every mode-$n$ fiber of the tensor $\mathcal{X}$. Tucker decomposition is a fundamental higher-order generalization of SVD. It decomposes a tensor $\mathcal{X}$ into a (typically smaller) core tensor $\mathcal{G} \in \mathbb{R}^{R_1 \times \cdots \times R_N}$ and a set of factor matrices $\{U^{(n)} \in \mathbb{R}^{I_n \times R_n}\}_{n=1}^N$, one for each mode. The decomposition is expressed as $\mathcal{X} \approx \mathcal{G} \times_1 U^{(1)} \times_2 U^{(2)} \cdots \times_N U^{(N)}$. The core tensor $\mathcal{G}$ can be interpreted as containing the latent interactions between the factors, while the factor matrices $U^{(n)}$ represent the principal components in each mode. The dimensions of the core tensor, $(R_1, ..., R_N)$, are referred to as the Tucker ranks of the decomposition. By choosing $R_n \ll I_n$, a compressed representation of $\mathcal{X}$ is achieved.

## 3 DESIGN OF LeSTD

LeSTD is a two-stage, data-free post-training compression framework for the MHA blocks of LLMs. Its goal is to dramatically reduce the storage cost of MHA parameters while preserving model accuracy. Figure 3 provides an overview of LeSTD. At a high level, LeSTD first learns a shared low-rank orthonormal subspace in which all heads and all $Q/K/V/O$ projections can be represented compactly (Stage I), and then sparsifies the resulting core tensor in this subspace with closed-form error control (Stage II). The final model performs inference directly in the compressed domain, without ever reconstructing the original dense weight matrices.

**Two-stage design.** In Stage I (refer to Section 3.1), we perform an iterative, shared-subspace Tucker decomposition (left sides of Figure 2 and Figure 3) on the MHA weight tensor. This procedure yields a set of shared orthogonal factor matrices $U^{(n)}$ and, for each attention head $i$, a small but dense core tensor $\mathcal{G}_i$. By iteratively refining the shared factors to jointly represent all heads, Stage I captures the common structure across heads while allowing head-specific variation to reside in the cores. In Stage II (refer to Section 3.2), we explicitly address the remaining dense-core bottleneck by introducing an importance-based pruning algorithm that learns an ultra-sparse core tensor. By systematically identifying and removing the least important core coefficients, and refitting the remaining ones in closed form, Stage II maximizes sparsity while tightly controlling the reconstruction error.

Both stages are executed sequentially (Stage I followed by Stage II), resulting in a highly compact representation of the MHA block. Section 3.3 then describes how the compressed model carries out inference without reconstructing the original weight matrices, thereby reducing storage and transmission costs: all $Q/K/V/O$ projections are evaluated directly from the shared factors and the sparse core, i.e., LeSTD enables LLMs to run inference directly on compressed weights without any decompression step.

## 3.1 SHARED-SUBSPACE TUCKER DECOMPOSITION (STAGE I)

**Tensorizing MHA Parameters.** Following the recent works (Gu et al., 2025), we restructure MHA parameters into a single 4th-order tensor (as Figure 3 Left). For each attention head $i$, we stack it four weight matrices: $\boldsymbol{W}_i^Q, \boldsymbol{W}_i^K, \boldsymbol{W}_i^V \in \mathbb{R}^{d_{\text{model}} \times d_{\text{head}}}$ where $d_{\text{head}} = d_{\text{model}}/h$, and $(\boldsymbol{W}_i^O)^\top \in \mathbb{R}^{d_{\text{model}} \times d_{\text{head}}}$, into a 3D tensor: $\boldsymbol{\mathcal{W}}_i = \left[\boldsymbol{W}_i^Q, \boldsymbol{W}_i^K, \boldsymbol{W}_i^V, (\boldsymbol{W}_i^O)^\top\right] \in \mathbb{R}^{d_{\text{model}} \times d_{\text{head}} \times 4}$. We then stack these $h$ tensors along a fourth mode to form the final MHA weight tensor: $\boldsymbol{\mathcal{W}}_{\text{total}} \in \mathbb{R}^{d_{\text{model}} \times d_{\text{head}} \times 4 \times h}$.
**Shared Low-Rank Decomposition.** We perform a standard Tucker decomposition (Malik & Becker, 2018; Ahmadi-Asl et al., 2021) of $\boldsymbol{\mathcal{W}}_{\text{total}}$ that factorizes along modes 1, 2, and 3, while leaving mode-4 (the head index) uncompressed. This means all heads share the same factor matrices ($\boldsymbol{U}^{(n)}, n = \{1, 2, 3\}$) but each retains its own slice of the core tensor ($\boldsymbol{\mathcal{G}}_i, i \in \{1, 2, ..., h\}$). Topology of decomposited $\boldsymbol{\mathcal{W}}_{\text{total}}$ is shown in Figure 4. Approximation is:

$$\boldsymbol{\mathcal{W}}_{\text{total}} \approx \hat{\boldsymbol{\mathcal{W}}}_{\text{total}} = \boldsymbol{\mathcal{G}}_{\text{total}} \times_1 \boldsymbol{U}^{(1)} \times_2 \boldsymbol{U}^{(2)} \times_3 \boldsymbol{U}^{(3)} \tag{3}$$

where $\boldsymbol{\mathcal{G}}_{\text{total}} \in \mathbb{R}^{R_1 \times R_2 \times R_3 \times h}$ is the dense core tensor produced in Stage I, and $\boldsymbol{U}^{(1)}, \boldsymbol{U}^{(2)}, \boldsymbol{U}^{(3)}$ are the shared column-orthogonal factor matrices with column dimensions $R_1, R_2, R_3$ respectively (Tucker ranks ($R_n \ll I_n$). Since we do not factorize the head mode, each head $i$ retains its own core slice $\boldsymbol{\mathcal{G}}_i = \boldsymbol{\mathcal{G}}_{\text{total}}[:, :, :, i] \in \mathbb{R}^{R_1 \times R_2 \times R_3}$ that can encode head-specific information, yet all slice $\boldsymbol{\mathcal{G}}_i$ live in joint coordinate system defined by the shared factors $\boldsymbol{U}^{(1)}, \boldsymbol{U}^{(2)}, \boldsymbol{U}^{(3)}$.
**Optimization Procedure.** We determine $\boldsymbol{U}^{(1)}, \boldsymbol{U}^{(2)}, \boldsymbol{U}^{(3)}$ and the $\boldsymbol{\mathcal{G}}_{\text{total}}$ by minimizing reconstruction error between the original and factorized weights (Böttcher & Wenzel, 2008):

$$\min_{\boldsymbol{U}^{(1)}, \boldsymbol{U}^{(2)}, \boldsymbol{U}^{(3)}, \boldsymbol{\mathcal{G}}_{\text{total}}} \|\boldsymbol{\mathcal{W}}_{\text{total}} - \hat{\boldsymbol{\mathcal{W}}}_{\text{total}}\|_F^2 \quad \text{s.t.} \quad (\boldsymbol{U}^{(n)})^\top \boldsymbol{U}^{(n)} = \boldsymbol{I}_{R_n} \tag{4}$$

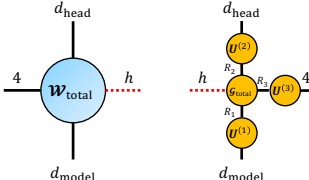

We solve Eq. (4) using the Higher-Order Orthogonal Iteration (HOOI) algorithm (Liu et al., 2014; Matstoms, 1997). Each sweep updates one factor by projecting $\boldsymbol{\mathcal{W}}_{\text{total}}$ onto the current subspaces of the other modes, taking a truncated SVD of the resulting mode-$n$ matricization, and setting $\boldsymbol{U}^{(n)}$ to the top-$R_n$ left singular vectors. After updating modes $1, 2, 3$, the core is recomputed as

Figure 4: The shape of $\boldsymbol{\mathcal{W}}_{\text{total}}$ before (Left) and after (Right) decomposition.

$\boldsymbol{\mathcal{G}}_{\text{total}} = \boldsymbol{\mathcal{W}}_{\text{total}} \times_1 (\boldsymbol{U}^{(1)})^\top \times_2 (\boldsymbol{U}^{(2)})^\top \times_3 (\boldsymbol{U}^{(3)})^\top$. This loop repeats until convergence, yielding the shared low-rank factors and the dense core tensor for all heads.

## 3.2 LEARNING AN ULTRA-SPARSE CORE VIA IMPORTANCE-BASED PRUNING (STAGE II)

Stage I (Eq. (3) in Section 3.1) yields a high-quality, but dense core tensor $\boldsymbol{\mathcal{G}}_{\text{total}}$ within a shared orthonormal subspace. We now aim to eliminate the dense core bottleneck by enforcing extreme sparsity on the $\boldsymbol{\mathcal{G}}_{\text{total}}$ while preserving reconstruction fidelity. Our approach in Stage II is explicitly importance-driven: we prune the least important elements of $\boldsymbol{\mathcal{G}}_{\text{total}}$ and set them to zero, thereby producing an ultra-sparse core $\boldsymbol{\mathcal{G}}_{\text{sparse}}$ that maintains the accuracy under a shared orthonormal (Stage I). Overall procedure of Stage II is presented in Algorithm 1. The critical question is thus: how should we define and compute the importance of an individual core

---

**Algorithm 1:** Stage II - Importance-based Core Pruning

**Input** : $\boldsymbol{\mathcal{G}}_{\text{total}}, \boldsymbol{U}^{(n)}(n = 1, 2, 3), \boldsymbol{\mathcal{W}}_{\text{total}}$, target sparsity $S_{\text{target}}$, pruning rate $\alpha$
**Output:** $\boldsymbol{\mathcal{G}}_{\text{sparse}}$
Initialize $\boldsymbol{\mathcal{G}} \leftarrow \boldsymbol{\mathcal{G}}_{\text{total}}$
**while** $\|\boldsymbol{\mathcal{G}}\|_0 > S_{\text{target}}$ **do**
    # Step 1: Compute Importance
    Calculate $\text{Imp}(g_\beta)$ for all non-zero $g_\beta \in \boldsymbol{\mathcal{G}}$ using Eq. (5)
    # Step 2: Prune
    $k \leftarrow \|\boldsymbol{\mathcal{G}}\|_0$ // number of non-zero elements
    $\Lambda \leftarrow \lceil \alpha \cdot k \rceil$       // lowest $\text{Imp}(\cdot)$ score
    **for** $\beta \in \Lambda$ **do**
        $g_\beta \leftarrow 0$ mark as pruned
    # Step 3: Refit
    **for** *each unpruned element* $g_\gamma \in \boldsymbol{\mathcal{G}}$ **do**
        Update $g_\gamma$ using Eq. (6)

**return** $\boldsymbol{\mathcal{G}}_{\text{sparse}} \leftarrow \boldsymbol{\mathcal{G}}$

---

element? We work entirely within the orthonormal subspace identified in Stage I (Eq. (3)). Recall that Stage I reconstruction is $\hat{\mathcal{W}}_{\text{total}} = \mathcal{G}_{\text{total}} \times_1 U^{(1)} \times_2 U^{(2)} \times_3 U^{(3)}$. Let $\beta = (r_1, r_2, r_3, i)$ be a multi-index and denote by $u_{r_n}^{(n)}$ the $r_n$-th column of $U^{(n)}$. Define rank-1 basis tensor associated with $\beta$ as the outer product: $\mathcal{B}_\beta = u_{r_1}^{(1)} \circ u_{r_2}^{(2)} \circ u_{r_3}^{(3)} \circ e_i$ where $e_i$ is the $i$-th standard basis vector along the head mode. Because factor matrices are column-orthonormal, the collection $\{\mathcal{B}_\beta\}$ is mutually orthonormal with unit Frobenius norms, i.e., $\langle \mathcal{B}_\beta, \mathcal{B}_\gamma \rangle = \delta_{\beta\gamma}$ and $\|\mathcal{B}_\beta\|_F = 1$. Let $\mathcal{R} = \mathcal{W}_{\text{total}} - \hat{\mathcal{W}}_{\text{total}}$ be Stage I residual. Since $\hat{\mathcal{W}}_{\text{total}}$ is the orthogonal projection of $\mathcal{W}_{\text{total}}$ onto the span of $\{\mathcal{B}_\beta\}$, orthogonality of the decomposition implies that: $\langle \mathcal{R}, \mathcal{B}_\beta \rangle = 0$ and $\|\mathcal{W}_{\text{total}}\|_F^2 = \|\mathcal{R}\|_F^2 + \sum_\beta g_\beta^2$, where $g_\beta$ denotes the coefficient of $\mathcal{B}_\beta$ in $\mathcal{G}_{\text{total}}$.

**Defining Importance.** We quantify the importance of an element $g_\beta$ of the normalized increase in reconstruction error caused by zeroing it out while holding all other quantities fixed. Writing Stage I error as $\mathcal{E} = \|\mathcal{R}\|_F^2$, consider the hypothetical reconstruction in which $g_\beta$ is set to zero, the corresponding reconstruction tensor is $\hat{\mathcal{W}}_{\text{total}}(g_\beta=0) = \hat{\mathcal{W}}_{\text{total}} - g_\beta \mathcal{B}_\beta$, because removing a single basis coefficient subtracts exactly its rank-1 contribution. The new residual is therefore

$$\mathcal{R}(g_\beta=0) = \mathcal{W}_{\text{total}} - \hat{\mathcal{W}}_{\text{total}}(g_\beta=0) = \mathcal{R} + g_\beta \mathcal{B}_\beta$$

Corresponding reconstruction error is obtained by expanding Frobenius norm and the orthogonality:

$$\mathcal{E}(g_\beta=0) = \|\mathcal{R} + g_\beta \mathcal{B}_\beta\|_F^2 = \langle \mathcal{R} + g_\beta \mathcal{B}_\beta \mathcal{R} + g_\beta \mathcal{B}_\beta \rangle = \|\mathcal{R}\|_F^2 + 2g_\beta \langle \mathcal{R}, \mathcal{B}_\beta \rangle + g_\beta^2 \|\mathcal{B}_\beta\|_F^2$$
$$= \mathcal{E} + 2g_\beta \cdot \mathbf{0} + g_\beta^2 \cdot 1 = \mathcal{E} + g_\beta^2$$

which yields the efficient expression (as shown in Algorithm 1, Step 1):

$$\mathcal{E}(g_\beta = 0) = \mathcal{E} + g_\beta^2 \tag{5}$$

Hence the normalized importance is $\text{Imp}(g_\beta) = \frac{\mathcal{E}(g_\beta=0)-\mathcal{E}}{\mathcal{E}} = \frac{g_\beta^2}{\mathcal{E}}$. Because the basis tensors are orthonormal, ordering elements by $\text{Imp}(g_\beta)$ is exactly equivalent to ordering by $|g_\beta|$. It follows that, under the Frobenius loss, the best $k$-term approximation is achieved by keeping the $k$ largest $|g_\beta|$ coefficients (i.e., hard-thresholding). This gives a principled, closed-form criterion for pruning: the least important elements (smallest magnitudes) are removed first and set to 0 (Algorithm 1 Step 2).

**Refitting remaining coefficients.** After pruning a subset of indices $\Lambda$ (setting $g_\beta \leftarrow 0$ for $\beta \in \Lambda$), we optionally refit remaining coefficients to counter numerical drift (Step 3). Fix all factors and the pruned pattern, and consider a surviving index $\gamma \notin \Lambda$. We determine the optimal $g_\gamma$ by solving one-dimensional least-squares problem: $g_\gamma^\star \leftarrow \arg\min_{g_\gamma'} \|\mathcal{W}_{\text{total}} - (g_\gamma' \mathcal{B}_\gamma + \sum_{\beta \neq \gamma \beta \notin \Lambda} g_\beta \mathcal{B}_\beta)\|_F^2$. Expanding the squared norm and differentiating with respect to $g_\gamma'$, we obtain the first-order optimality condition: $\frac{\partial}{\partial g_\gamma'}\left(\|\mathcal{W}_{\text{total}}\|_F^2 - 2g_\gamma' \langle \mathcal{W}_{\text{total}}, \mathcal{B}_\gamma \rangle + 2g_\gamma' \sum_{\beta \neq \gamma, \beta \notin \Lambda} g_\beta \langle \mathcal{B}_\beta, \mathcal{B}_\gamma \rangle + (g_\gamma')^2 \|\mathcal{B}_\gamma\|_F^2\right) = 0$. Using $\langle \mathcal{B}_\beta, \mathcal{B}_\gamma \rangle = \delta_{\beta\gamma}$ and $\|\mathcal{B}_\gamma\|_F^2 = 1$, the mixed terms vanish and we get $-2\langle \mathcal{W}_{\text{total}}, \mathcal{B}_\gamma \rangle + 2g_\gamma' = 0 \Rightarrow g_\gamma' = \langle \mathcal{W}_{\text{total}}, \mathcal{B}_\gamma \rangle$. Thus the refitting update takes the closed form:

$$g_\gamma \leftarrow \langle \mathcal{W}_{\text{total}}, \mathcal{B}_\gamma \rangle \tag{6}$$

which coincides with original coefficient when the factors are fixed and arithmetic is exact. Performing this refit after each pruning step helps to stabilize the reconstruction in finite precision.

### 3.3 INFERENCE WITHOUT RECONSTRUCTION

A key benefit of LeSTD is that inference operates directly in the compressed domain: the full dense matrices are never materialized. All computations are expressed through the shared low-rank factors (in Stage I) and the learned sparse core (in Stage II). The overall process is shown in Algorithm 2, we detailed the steps below:

**One-off left projection (shared across heads).** Let $X \in \mathbb{R}^{T \times d_{\text{model}}}$ be token representations (embeddings) for a mini-batch (or $T = 1$ in the autoregressive single-step). We project $X$ once into the rank-$R_1$ subspace and reuse it for all heads and all of $Q, K, V$: $Y = X U^{(1)} \in \mathbb{R}^{T \times R_1}$.

**Core contraction and per-head slices.** Denote Stage II sparse core by $\mathcal{G}_{\text{sparse}} \in \mathbb{R}^{R_1 \times R_2 \times R_3 \times h}$ and the third-mode factor by $U^{(3)} \in \mathbb{R}^{4 \times R_3}$, where $t \in \{1{:}Q, 2{:}K, 3{:}V, 4{:}O\}$ indexes the projection type. For each head $i$ and type $t$, we define the slice, which is also typically sparse due to $\mathcal{G}_{\text{sparse}}$, as: $M_{i,t}[r_1, r_2] = \sum_{r_3=1}^{R_3} \mathcal{G}_{\text{sparse}}[r_1, r_2, r_3, i] \cdot U^{(3)}[t, r_3]$, where $M_{i,t}[r_1, r_2] \in \mathbb{R}^{R_1 \times R_2}$ and depends

only on the compressed parameters and can be precomputed and cached offline.

**Input-side linear maps $(Q, K, V)$ without reconstruction.** With $\boldsymbol{U}^{(2)} \in \mathbb{R}^{d_{\text{head}} \times R_2}$, the three projections (Eq. (1)) for head $i$ are computed by $\boldsymbol{Q}_i = \boldsymbol{Y} \boldsymbol{M}_{i,1}(\boldsymbol{U}^{(2)})^\top$, $\boldsymbol{K}_i = \boldsymbol{Y} \boldsymbol{M}_{i,2}(\boldsymbol{U}^{(2)})^\top$, and $\boldsymbol{V}_i = \boldsymbol{Y} \boldsymbol{M}_{i,3}(\boldsymbol{U}^{(2)})^\top$, yielding $\boldsymbol{Q}_i, \boldsymbol{K}_i, \boldsymbol{V}_i \in \mathbb{R}^{T \times d_{\text{head}}}$ without ever forming $\boldsymbol{W}_i^Q, \boldsymbol{W}_i^K, \boldsymbol{W}_i^V$.

**Attention and output-side map without reconstruction.** We scaled dot-product attention proceeds as usual: $\boldsymbol{A}_i = \text{softmax}\left(\frac{\boldsymbol{Q}_i \boldsymbol{K}_i^\top}{\sqrt{d_{\text{head}}}}\right)$ and $\boldsymbol{O}_i = \boldsymbol{A}_i \boldsymbol{V}_i \in \mathbb{R}^{T \times d_{\text{head}}}$. For output projection (Eq. (2)), note that $(\boldsymbol{W}_i^O)^\top = \boldsymbol{U}^{(1)} \boldsymbol{M}_{i,4}(\boldsymbol{U}^{(2)})^\top \Rightarrow \boldsymbol{W}_i^O = \boldsymbol{U}^{(2)} \boldsymbol{M}_{i,4}^\top (\boldsymbol{U}^{(1)})^\top$. Thus, projected output for head $i$ is $\boldsymbol{O}_i' = \boldsymbol{O}_i \boldsymbol{W}_i^O = (\boldsymbol{O}_i \boldsymbol{U}^{(2)}) \boldsymbol{M}_{i,4}^\top (\boldsymbol{U}^{(1)})^\top \in \mathbb{R}^{T \times d_{\text{model}}}$. Aggregating all attention heads (e.g., Concat in original Eq. (2) gives the final output. An optimization is to delay the final multiplication by $(\boldsymbol{U}^{(1)})^\top$: $\sum_{i=1}^h \boldsymbol{O}_i' = \left(\sum_{i=1}^h (\boldsymbol{O}_i \boldsymbol{U}^{(2)}) \boldsymbol{M}_{i,4}^\top\right)(\boldsymbol{U}^{(1)})^\top$, so that $(\boldsymbol{U}^{(1)})^\top$ is applied only once.

By performing above processes, the original matrices $\{\boldsymbol{W}_i^Q, \boldsymbol{W}_i^K, \boldsymbol{W}_i^V, \boldsymbol{W}_i^O\}$ are not instantiated. All products are composed from $\boldsymbol{U}^{(1)}, \boldsymbol{U}^{(2)}$, precomputed (sparse) $\boldsymbol{M}_{i,t}$, and activations. A formal theoretical complexity analysis is deferred to Appendix F.

---

**Algorithm 2:** Inference Without Reconstruction in LeSTD

---

**Input** : Token representations $\boldsymbol{X} \in \mathbb{R}^{T \times d_{\text{model}}}$; Shared factors $\boldsymbol{U}^{(1)} \in \mathbb{R}^{d_{\text{model}} \times R_1}$,
$\qquad\quad \boldsymbol{U}^{(2)} \in \mathbb{R}^{d_{\text{head}} \times R_2}$, $\boldsymbol{U}^{(3)} \in \mathbb{R}^{4 \times R_3}$; Sparse core $\boldsymbol{\mathcal{G}}_{\text{sparse}} \in \mathbb{R}^{R_1 \times R_2 \times R_3 \times h}$.

**Output:** MHA output $\boldsymbol{Z} \in \mathbb{R}^{T \times d_{\text{model}}}$.

\# One-off left projection (shared across heads and types)

$\boldsymbol{Y} \leftarrow \boldsymbol{X} \boldsymbol{U}^{(1)} \in \mathbb{R}^{T \times R_1}$

\# Precompute per-head, per-type compressed matrices

**for** $i \leftarrow 1$ **to** $h$ **do**
$\quad$ **for** $t \in \{1{:}Q, 2{:}K, 3{:}V, 4{:}O\}$ **do**
$\quad\quad$ **for** $r_1 \leftarrow 1$ **to** $R_1$ **do**
$\quad\quad\quad$ **for** $r_2 \leftarrow 1$ **to** $R_2$ **do**
$\quad\quad\quad\quad \boldsymbol{M}_{i,t}[r_1, r_2] \leftarrow \sum_{r_3=1}^{R_3} \boldsymbol{\mathcal{G}}_{\text{sparse}}[r_1, r_2, r_3, i]\, \boldsymbol{U}^{(3)}[t, r_3]$

\# Input-side linear maps $(Q, K, V)$ in compressed form

**for** $i \leftarrow 1$ **to** $h$ **do**
$\quad \boldsymbol{Q}_i \leftarrow \boldsymbol{Y}\, \boldsymbol{M}_{i,1}(\boldsymbol{U}^{(2)})^\top \in \mathbb{R}^{T \times d_{\text{head}}}$
$\quad \boldsymbol{K}_i \leftarrow \boldsymbol{Y}\, \boldsymbol{M}_{i,2}(\boldsymbol{U}^{(2)})^\top \in \mathbb{R}^{T \times d_{\text{head}}}$
$\quad \boldsymbol{V}_i \leftarrow \boldsymbol{Y}\, \boldsymbol{M}_{i,3}(\boldsymbol{U}^{(2)})^\top \in \mathbb{R}^{T \times d_{\text{head}}}$

\# Scaled dot-product attention

**for** $i \leftarrow 1$ **to** $h$ **do**
$\quad \boldsymbol{A}_i \leftarrow \text{softmax}\left(\frac{\boldsymbol{Q}_i \boldsymbol{K}_i^\top}{\sqrt{d_{\text{head}}}}\right)$
$\quad \boldsymbol{O}_i \leftarrow \boldsymbol{A}_i \boldsymbol{V}_i \in \mathbb{R}^{T \times d_{\text{head}}}$

\# Output-side projection in compressed form

**for** $i \leftarrow 1$ **to** $h$ **do**
$\quad \boldsymbol{H}_i \leftarrow \boldsymbol{O}_i \boldsymbol{U}^{(2)} \in \mathbb{R}^{T \times R_2}$
$\quad \boldsymbol{S}_i \leftarrow \boldsymbol{H}_i \boldsymbol{M}_{i,4}^\top \in \mathbb{R}^{T \times R_1}$

$\boldsymbol{S} \leftarrow \sum_{i=1}^h \boldsymbol{S}_i \in \mathbb{R}^{T \times R_1}$

\# Final projection back to model dimension

$\boldsymbol{Z} \leftarrow \boldsymbol{S}(\boldsymbol{U}^{(1)})^\top \in \mathbb{R}^{T \times d_{\text{model}}}$

**return** $\boldsymbol{Z}$

---

## 4 EVALUATION

We compare LeSTD against SVD-based approaches (Yuan et al., 2023; Wang et al., 2024b; Gu et al., 2025) and pruning techniques (Ashkboos et al., 2024). We further consider quantization as

another representative post-training compression paradigm (Huang et al., 2024; Shang et al., 2023). Specifically, we examine the direct comparisons in terms of compression ratio *vs.* model accuracy, with detailed results and analysis provided in Appendix D.

We conduct a series of internal ablation studies to validate LeSTD's design choices. These studies analysis the individual contributions of our two-stage approach and examine the model's sensitivity to key hyperparameters, such as Tucker ranks. The comprehensive results, which confirm effectiveness of each component, are presented in Appendix C.

## 4.1 EXPERIMENTAL SETTING

**Baselines.** We compare LeSTD with three post-training LLM compression schemes: (1) Low-rank decomposition: ASVD (Yuan et al., 2023), SVD-LLM (Wang et al., 2024b), and TensorLLM (Gu et al., 2025); (2) Pruning: SliceGPT (Ashkboos et al., 2024); (3) Quantization: PB-LLM (Shang et al., 2023) and BiLLM (Huang et al., 2024). Note that LeSTD is a post-training compression framework. Training-aware approaches, such as OneBit (Xu et al., 2024), fall outside scope of this work. Please refer to Appendix E for a detailed discussion of related works.
**LLM models and Datasets.** We evaluate performance of LeSTD and baselines on three families of LLM models at different scales (GPT-J (Wang, 2021), Llama2-13B (Touvron et al., 2023), and OPT-30B (Zhang et al., 2022)). Experiments spans four datasets covering diverse tasks: natural language modeling (WikiText-2 (Merity et al., 2016)), natural language understanding (MathQA (Amini et al., 2019)), math reasoning (GSM8K (Cobbe et al., 2021)), and the natural language generation (TruthfulQA (Lin et al., 2021)). For a detailed description of datasets, please refer to Appendix A.
**Software&Hardware Environment.** All experiments are conducted on a server equipped with an Intel(R) Xeon(R) Platinum 8470 CPU, 1024 GB of DDR4 memory, and one NVIDIA H100 GPU with 80 GB GPU memory, running Ubuntu 22.04.5 LTS. Implementation is based on Python 3.10.12 and PyTorch 2.7.0, with CUDA version of 12.9. The efficiency index (*tokens/sec*) were measured under the Transformer Library (Wolf et al., 2020) (version 4.53.0). For detailed setting of baselines and LeSTD, such as hyperparameters and implementation, please refer to Appendix B. Due to the presence of Stage II (Section 3.2), compression ratio of LeSTD is not as straightforward to compute as that of baselines. In Appendix G, we present our approach for precisely controlling the compression ratio budget in LeSTD.

## 4.2 OVERALL COMPARISON

We evaluate four datasets: WikiText-2, MathQA, GSM8K, and TruthfulQA across three LLMs with different scales/families, namely GPT-J (6B), Llama2-13B, and OPT-30B. Evaluation focuses on two aspects: trade-off between compression ratio and model accuracy, and inference efficiency, the latter quantified by tokens per second (*tokens/sec*).
**Compression Rate *vs.* The Model Accuracy.** As shown in the Table 1, across all the three model scales/families and four benchmarks, same trend recurs: as the compression strengthens, LeSTD preserves accuracy more faithfully than matrix-by-matrix SVD (ASVD and SVD-LLM) and than dense-core Tucker (TensorLLM). At a aggressive ratio (e.g., 0.2), LeSTD lowers WikiText-2 perplexity by $\sim 15\%$ relative to SVD-LLM (e.g., 42.70 vs. 50.25 on OPT-30B). Besides, when the core becomes the bottleneck for Tucker at very small ratio (e.g., 0.2), LeSTD outperforms TensorLLM by $\sim 10\%$ (e.g., 80.37 *vs.* 89.52 on GPT-J). On MathQA and GSM8K, LeSTD typically matches or edges out the best baseline by $\sim 3$ points, and on TruthfulQA it remains stable where the pruning scheme (SliceGPT) or matrix-by-matrix SVD can collapse.

These results align with our design and analysis. Stage I learns a single, shared subspace per layer (Eq.(3) and Eq.(4)) that captures cross-head regularities, explaining the flatter degradation relative to matrix-wise SVD. Stage II then breaks the dense-core ceiling by pruning in an orthonormal latent basis, where reconstruction is an orthogonal projection and coefficient importance has a closed form (Eq.(5) and Eq.(6)). This is why LeSTD sustains accuracy at compression levels where dense-core Tucker, for example, TensorLLM, must retain large ranks.
**Inference Efficiency.** As shown in the Figure 5, LeSTD delivers competitive-to-superior throughput while retaining accuracy across LLMs, datasets and compression rate. At a mild compression (e.g., 0.8), it is at least on par with fastest baselines and often ahead: on WikiText-2 it edges SVD-LLM on GPT-J (11571.82 *vs.* 11521.82) and improves $\sim 3\%$ on Llama2-13B. The gains over

Table 1: The overall results of compression rate (defined as $\frac{\text{compressed}}{\text{original}}$) *vs.* model accuracy across four datasets. Some results are marked as "–", indicating that value is 0 when rounded to two decimal places. Uparrow "↑" indicates higher measure value are better, and downarrow "↓" indicates lower is better. Columns with a compression ratio of 1 refers to the original model without compression.

(a) WikiText2 (Metric: Perplexity ↓)

| Ratio / Method | GPT-J | | | | | Llama2-13B | | | | | OPT-30B | | | | |
|---|---|---|---|---|---|---|---|---|---|---|---|---|---|---|---|
| | 0.2 | 0.4 | 0.6 | 0.8 | 1 | 0.2 | 0.4 | 0.6 | 0.8 | 1 | 0.2 | 0.4 | 0.6 | 0.8 | 1 |
| ASVD | 4412.47 | 3873.72 | 273.52 | 15.91 | | 2826.88 | 139.26 | 49.95 | 14.97 | | 2423.65 | 998.26 | 100.48 | 24.89 | |
| SVD-LLM | 100.08 | 50.11 | 20.19 | 14.95 | | 34.25 | 22.52 | 14.88 | 9.93 | | 50.25 | 20.09 | 14.34 | 11.95 | |
| TensorLLM | 89.52 | 49.70 | 9.90 | **8.92** | 8.86 | 14.97 | 14.09 | 9.96 | 8.01 | 4.88 | 49.53 | **19.97** | 14.92 | 11.95 | 9.56 |
| SliceGPT | 502.87 | 397.62 | 59.69 | 12.01 | | 50.09 | 19.83 | 14.97 | 10.05 | | 45.03 | 24.77 | 15.08 | 9.99 | |
| LeSTD (Ours) | **80.37** | **49.56** | **9.51** | **8.92** | | **13.99** | **11.94** | **7.93** | **6.98** | | **42.70** | 20.13 | **14.02** | **9.98** | |

(b) MathQA (Metric: Accuracy ↑)

| Ratio / Method | GPT-J | | | | | Llama2-13B | | | | | OPT-30B | | | | |
|---|---|---|---|---|---|---|---|---|---|---|---|---|---|---|---|
| | 0.2 | 0.4 | 0.6 | 0.8 | 1 | 0.2 | 0.4 | 0.6 | 0.8 | 1 | 0.2 | 0.4 | 0.6 | 0.8 | 1 |
| ASVD | 0.14 | **0.20** | **0.21** | **0.22** | | – | 0.08 | 0.11 | 0.26 | | – | 0.06 | 0.17 | 0.20 | |
| SVD-LLM | **0.18** | **0.20** | **0.21** | **0.22** | | 0.16 | 0.22 | 0.24 | **0.30** | | 0.14 | 0.21 | **0.26** | **0.29** | |
| TensorLLM | 0.16 | 0.19 | 0.20 | 0.21 | 0.23 | 0.15 | 0.20 | 0.26 | 0.29 | 0.32 | 0.10 | 0.18 | 0.24 | 0.28 | 0.31 |
| SliceGPT | 0.11 | 0.16 | 0.19 | 0.21 | | 0.06 | 0.14 | 0.24 | 0.27 | | 0.06 | 0.15 | 0.20 | 0.28 | |
| LeSTD (Ours) | 0.14 | 0.19 | 0.20 | **0.22** | | 0.16 | **0.23** | **0.26** | **0.30** | | 0.13 | **0.22** | 0.25 | **0.29** | |

(c) GSM8K (Metric: Exact-match Accuracy ↑)

| Ratio / Method | GPT-J | | | | | Llama2-13B | | | | | OPT-30B | | | | |
|---|---|---|---|---|---|---|---|---|---|---|---|---|---|---|---|
| | 0.2 | 0.4 | 0.6 | 0.8 | 1 | 0.2 | 0.4 | 0.6 | 0.8 | 1 | 0.2 | 0.4 | 0.6 | 0.8 | 1 |
| ASVD | – | – | 0.02 | 0.08 | | – | – | 0.06 | 0.16 | | – | – | 0.03 | 0.07 | |
| SVD-LLM | – | – | – | 0.03 | | 0.06 | 0.10 | 0.14 | 0.20 | | 0.03 | 0.04 | **0.08** | **0.10** | |
| TensorLLM | – | 0.01 | 0.03 | 0.05 | 0.13 | 0.03 | 0.08 | 0.14 | 0.19 | 0.23 | 0.02 | 0.03 | 0.07 | 0.08 | 0.11 |
| SliceGPT | – | – | 0.01 | 0.04 | | – | 0.03 | **0.17** | **0.21** | | – | 0.02 | 0.05 | 0.08 | |
| LeSTD | – | 0.02 | 0.04 | **0.09** | | 0.05 | 0.08 | **0.17** | **0.21** | | 0.03 | 0.03 | 0.07 | 0.09 | |

(d) TruthfulQA (Metric: BLEU-1 ↑)

| Ratio / Method | GPT-J | | | | | Llama2-13B | | | | | OPT-30B | | | | |
|---|---|---|---|---|---|---|---|---|---|---|---|---|---|---|---|
| | 0.2 | 0.4 | 0.6 | 0.8 | 1 | 0.2 | 0.4 | 0.6 | 0.8 | 1 | 0.2 | 0.4 | 0.6 | 0.8 | 1 |
| ASVD | – | – | – | 0.02 | | – | – | 0.03 | 0.05 | | – | – | 0.03 | 0.04 | |
| SVD-LLM | – | 0.01 | 0.01 | 0.05 | | 0.03 | 0.03 | 0.04 | 0.06 | | 0.03 | 0.03 | **0.06** | 0.07 | |
| TensorLLM | – | 0.02 | 0.02 | 0.04 | 0.06 | 0.02 | 0.02 | 0.05 | **0.07** | 0.08 | 0.02 | 0.03 | 0.05 | 0.07 | 0.08 |
| SliceGPT | – | – | 0.01 | 0.03 | | – | 0.02 | 0.03 | 0.06 | | – | – | 0.02 | 0.05 | |
| LeSTD | – | 0.02 | 0.04 | **0.06** | | 0.03 | 0.05 | **0.06** | **0.07** | | 0.01 | 0.03 | 0.05 | **0.08** | |

TensorLLM (dense-core Tucker) are large: up to $\sim 40\%$ improvement *tokens/sec*. For short-context workloads (Zhou et al., 2025) such as GSM8K and TruthfulQA, LeSTD's advantage at an aggressive compression (e.g., 0.2) is consistent, yielding $\sim 10\%$ improvement averagely over SVD-LLM on GPT-J and OPT-30B. On MathQA, throughput at 0.8 is mixed: small positive on GPT-J, near parity on Llama2-13B, and a modest deficit on OPT-30B, consistent with cases where heavily tuned GEMMs (General Matrix Multiplications) favor purely low-rank baselines (Sun et al., 2025).

At stronger compression (e.g., 0.4), LeSTD maintains competitive throughput as dense-core Tucker degrades more sharply and matrix-by-matrix SVD increasingly pays for redundant per-head projections. This behavior follows from LeSTD execution (Section 3.3): inference never reconstructs dense $Q, K, V, O$, a single shared left projection $\boldsymbol{X}\boldsymbol{U}^{(1)}$ is reused across heads and across $Q, K, V$, and the right side applies $(\boldsymbol{U}^{(2)})^\top$ with head-local, precomputable slices $\boldsymbol{M}_{i,t}$. Stage II's ultra-sparse core reduces arithmetic consumption, whereas dense-core Tucker (TensorLLM) retains polynomial many latent interactions. In summary, share once, sparsity what remains, and execute directly in compressed domain, the observed *tokens/sec* reflect exactly this design. To ensure our results reflect the real-world scenarios, all throughput benchmarks were conducted within the standard library (Wolf et al., 2020) (v4.53.0) without any custom-written sparse kernels. Observed speedups (Figure 5) stem from the library's underlying Pytorch (Paszke et al., 2019) implementation, which can leverage the reduced computational load from the LeSTD's sparse core tensor (Section 3.2) and inference on compressed (Section 3.3). While performance of unstructured sparsity can be sensitive to implementation and hardware details (Dave et al., 2021; Wang, 2020), our results demonstrate that LeSTD provides tangible wall-clock time improvements in a widely-used environment.

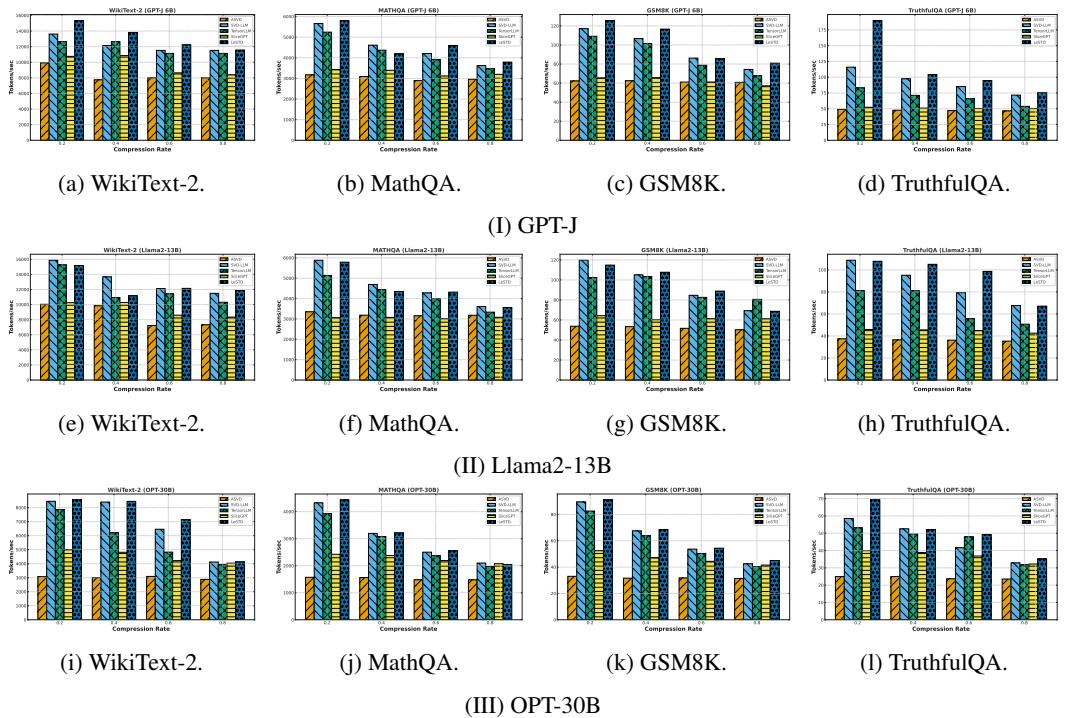

(a) WikiText-2.  (b) MathQA.  (c) GSM8K.  (d) TruthfulQA.

(I) GPT-J

(e) WikiText-2.  (f) MathQA.  (g) GSM8K.  (h) TruthfulQA.

(II) Llama2-13B

(i) WikiText-2.  (j) MathQA.  (k) GSM8K.  (l) TruthfulQA.

(III) OPT-30B

Figure 5: Throughput (*tokens/sec*) of LLMs across datasets, comparing baselines with our LeSTD

## 5 RELATED WORK

Recent research has explored post-training compression techniques for LLMs (Zhu et al., 2024). ASVD (Yuan et al., 2023) introduces activation-aware scaling to improve SVD-based low-rank approximations without retraining whole model. SVD-LLM (Wang et al., 2024b) further refines this by incorporating truncation-aware whitening and closed-form updates, achieving superior accuracy under aggressive compression. TensorLLM (Gu et al., 2025) applies Tucker decomposition to MHA, enforcing a shared subspace across heads for up to 250X compression. SliceGPT (Ashkboos et al., 2024) adopts a structured pruning approach by deleting rows and columns of weight matrices, reducing compute and memory while maintaining nearly full zero-shot accuracy. Our method, LeSTD, follows this post-training paradigm but departs from prior designs through its shared-subspace sparse Tucker formulation. A detailed description of related works, please refer to Appendix E.

## 6 CONCLUSION

We introduced LeSTD, a learning-based sparse tensor decomposition framework designed to solve the critical dense core bottleneck in tensor-based post-training LLM compression. The key point is a principled, two-stage design, which decouples basis optimization from core sparsification. This allows LeSTD to first learn a high-quality shared latent space for all attention heads, and then aggressively yet safely sparsify the core tensor within that stable basis. It is this ability to eliminate redundancy within the compressed latent space itself that allows LeSTD to break the compression ceiling imposed by dense-core methods. Our extensive experiments confirm this powerful synergy, showing that LeSTD consistently outperforms existing methods and establishes a new state of the art in the compression–accuracy trade-off for large language models.

## ACKNOWLEDGMENT

This work was partially supported by NSF 2343863, 2413520, 2417747 and 2440611. Any opinions, conclusions, or recommendations expressed in this material are those of the authors and do not necessarily reflect the views of the NSF.

LIMITATION

A limitation of LeSTD is that Stage II (Section 3.2) pruning produces an unstructured sparse core tensor. While this approach is highly effective at reducing theoretical FLOPs and storage (refer to Appendix F), leading to practical speedups we demonstrate using standard software libraries (Section 4.2), its performance on specialized hardware could be further amplified. Exploring methods to enforce structured sparsity (e.g., block-sparsity) within the core is a promising direction for better hardware co-design.

ETHICS STATEMENT

In this paper, we present a design grounded in sparse tensor factorization that addresses the growing scale of Large Language Models (LLMs). Our approach significantly reduces the storage and computational requirements of these models without compromising performance, thereby promoting the democratization and wider accessibility of powerful language technologies. This improvement holds potential for a wide range of applications by enabling the deployment of LLMs on resource-constrained devices. We believe that our method contributes positively to the advancement of machine learning research by promoting computational efficiency and sustainability. Although we do not anticipate any immediate negative ethical implications or societal concerns from our approach, it's important to acknowledge that machine learning technologies, including LLMs, have broader impacts. The increased accessibility of these models underscores the need for ongoing research into their potential for misuse and the development of robust safeguards. Therefore, responsible implementation and continued dialogue are crucial to ensure that such technologies are applied in a manner that promotes fairness, transparency, and beneficial societal outcomes.

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

## A DATASET

**WikiText-2 (Metric: Perplexity ↓).** A word-level language-modeling corpus built from Wikipedia "Good/Featured" articles that preserves case, punctuation, and numbers. The standard split contains 2.09M tokens (train), 217.6k (validation), and 245.6k (test) with a 33,278-word vocabulary, we evaluate by perplexity (lower is better).
**MathQA (Metric: Accuracy ↑).** A multiple-choice math word-problem dataset ($\approx$37k items) created by annotating AQuA problems with operation-programs. Authors use an 80/12/8 random split, i.e., $\approx$29.8k train/4.4k validation/3k test, and we report answer accuracy on the evaluation set.
**GSM8K (Metric: Accuracy ↑).** A grade-school math benchmark of 8.5k authored problems targeting multi-step reasoning. Canonical split is 7.5k train / 1k test. We follow the original protocol and report exact-match accuracy of the final numeric answer.
**TruthfulQA (Metric: BLEU-1 ↑).** A truthfulness benchmark of 817 questions across 38 categories (e.g., health, law, finance, politics). There is no official train/val/test split, evaluation is conducted on the full question set via multiple-choice and/or free-generation protocols. In our setup, we compute BLEU-1 (unigram BLEU) between model outputs and the provided references.

## B BASELINES AND LeSTD DETAILED SETTING

ASVD decomposes the attention weights with activation-aware scaling and a layerwise rank search, in the original paper, auhtor experiments use a small, fixed calibration corpus drawn from WikiText-2: 32 calibration datasets each containing 2048 tokens. We keep the exact same calibration budget, and choose per-layer truncation ranks to meet the desired global compression ratio (e.g., 0.2 - 0.8).

SVD-LLM introduces truncation-aware whitening plus sequential low-rank updates (SLRA). Its implementation details specify that whitening/truncation are fit on a random set of sentences with a concrete budge of 256 samples. SLRA recovery is performed with instruction data (Alpaca 50K (Taori et al., 2023)). In our experiments, we use 256 calibration sentences (from WikiText-2) with sequence length 2028 for whitening and truncation, and enable SLRA with SVD-LLM's defaults.

SliceGPT performed column slicing guided by a second-order/Taylor criterion (Li et al., 2017) and calibrates on either WikiText-2 or Alpaca. The paper fixes the main experiment calibration to 1024 sequences of length 2048, and uses the LM-Eval Harness (Gao et al., 2021) defaults for the zero-shot tasks. Slicing levels reported across models are 10-30% with optional LoRA-based recovery fine-tuning (with $r = 32, \alpha = 10$). To remain the faithful across MathQA, GSM8K and TruthfulQA (not originally supported in the paper), we keep the same calibration budget (1024×2048 on WikiText-2), set a target slicing fraction that match the compression grid ([0.2, 0.4, 0.6, 0.8] overall by solving for per-layer variance removal as in the paper), and evaluate with LM-Eval defaults.

TensorLLM factorizes multi-head attention (MHA) with a shared-factor Tucker decomposition and is explicitly post-training, requiring no extra data, calibration, or fine-tuning (Hu et al., 2022). The only operative knobs are the multi-linear ranks (equivalently a per-layer MHA compression rate), which the paper sweeps broadly (e.g., on GPT-J, reported the MHA compression rates span $\approx$14X - 247X depending on the layer/dataset) while keeping standard evaluation. For all four datasets, we reuse our dense prompts and simply set Tucker ranks to hit the target global compression ratios on the attetion block (start with 8 - 16X on $Q, K, V$ and 2 - 6X on output projections for decoder-only models, then adjust to match the overall ratio).

For two 1-bit quantization baselines (for experimental results, please refer to Appendix D), BiLLM and PB-LLM, emphasize ligh-weight post-training calibration on generic text. BiLLM quantize using C4 (Bandari et al., 2024) as the calibration source with blocksize 128 and the Hessian criterion for salient-weight selection. PB-LLM partially binarizes weights while preserving a small salient subset and use GPTQ-style (Frantar et al., 2022) Hessian-guided reconstruction. In our experiments, we use PTQ-only settings across four datasets, the choice is to mirror BiLLM: take 128 - 512 calibration sequences from C4 (length is 2048), use blocksize 128 and Hessian saliency for BiLLM, and for PB-LLM retain $\sim 5 - 10\%$ salient weights in higher precision with GPTQ-style Hessian reconstruction, and keep dataset prompts at LM-Eval defaults.

For our proposed LeSTD, we follow same evaluation setup as described above, but note that LeSTD is a fully data-free post-training method (no calibration data or retraining is used). We set the multi-

linear Tucker ranks $(R_1, R_2, R_3)$ in each attention layer so that the overall model meets the target compression ratio (0.2, 0.4, 0.6, 0.8). Concretely, for each desired global size fraction, we solve for the layer-wise ranks that yield approximately that compression of the MHA block. The Stage I (Section 3.1, shared Tucker factorization) is computed with HOOI using a fixed number of iterations (we typically run 5-10 iterations per layer, which reliably converges to a low reconstruction error). After Stage I, Stage II (Section 3.2) performs importance-baseed pruning on the core tensor as follows: we fix a per-iteration prune fraction $\alpha = 0.5$ (i.e., remove 10% of the smallest-magnitude core elements each round). We continue pruning iteratively until the core has reached the target sparsity (chosen so that the final compressed size matches the target ratio). After each pruning step, we refit the remaining core coefficients by the closed-form update (Eq.(6)) to counteract numerical drift, this ensures optimal reconstruction with the fixed basis. In summary, LeSTD requires no fine-tuning or task-specific data. All hyperparameters (Tucker ranks, pruning schedule) are determined by the global size target. Once the compression is done, inference proceeds directly on the compressed representation as described. In our experiments, we found that this setting reliably achieves the desired compression at the indicated performance levels, without using any extra data beyond what the evaluation tasks provide. Please also refer to Appendix G for a detailed explanation of how the compression ratio of LeSTD is precisely computed.

## C ABLATION STUDIES

### C.1 RANK-SENSITIVITY

To gauge the sensitivity to Tucker ranks $(R_1, R_2, R_3 = 4)$, we target an overall MHA compression ratio around 0.6 and vary $(R_1, R_2)$ while keeping $R_3$ fixed (since rank $R_3$ represents the $Q, K, V, O$ axis of size 4). Table 2 reports several representative configurations (the "Ratio" column shows the actual compression fraction achieved), and perplexity is measured on WikiText-2 with GPT-J (6B). This is not an exhaustive grid search, but it already covers a reasonably wide range of $(R_1, R_2)$ in the $\sim 0.6$ regime and illustrates clear trends: larger ranks generally yield lower perplexity. In particular, increasing $R_1$ or $R_2$ consistently reduces perplexity, and configurations such as $(R_1 = 2048, R_2 = 256, R_3 = 4)$ and $(R_1 = 2048, R_2 = 288, R_3 = 4)$ achieve near-original

Table 2: Perplexity under varying Tucker ranks for GPT-J (6B) at $\sim 0.6$ overall compression.

| R1 | R2 | R3 | Ratio | Perplexity ($\downarrow$) |
|---|---|---|---|---|
| 4096 | 256 | 4 | 1 | 8.86 |
| 1536 | 320 | 4 | 0.56 | 13.72 |
| 1536 | 384 | 4 | 0.66 | 11.91 |
| 1792 | 320 | 4 | 0.65 | 11.33 |
| 2048 | 224 | 4 | 0.56 | 10.53 |
| 2048 | 256 | 4 | 0.63 | 9.53 |
| 1920 | 256 | 4 | 0.60 | 9.92 |
| 2048 | 288 | 4 | 0.66 | 9.26 |

perplexity (down to 9.26) at compression ratios around 0.63–0.66, whereas smaller ranks (e.g., $R_1 = 1536, R_2 = 320, R_3 = 4$) lead to noticeable degradation. Overall, the best perplexity in this sweep occurs at higher ranks near the 0.6 compression point, confirming that our reported setting (Table 1) lies in a regime that balances model size and accuracy.

### C.2 EFFECT OF STAGE-II ON CORE STRUCTURE

We adopt a shared-subspace Tucker decomposition with orthonormal factors $\boldsymbol{U}^{(1)}, \boldsymbol{U}^{(2)}, \boldsymbol{U}^{(3)}$ along the corresponding modes, respectively, while leaving the head mode unfactorized. For display, as shown in Figure 6, we keep $R_3$ $(Q, K, V, O)$ fixed and a particular head $i = 5$, and show the resulting $R_1 \times R_2$ core slice. Both panels share a diverging colormap centered at zero for strict comparability. Stage II performs best-$k$ pruning in the same

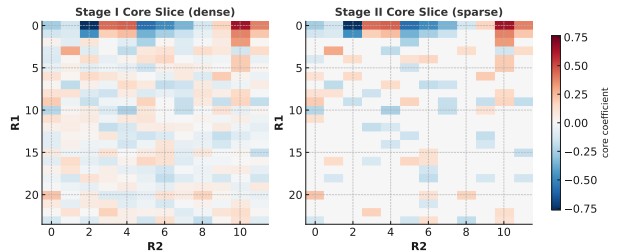

Figure 6: Visualization of a fixed Tucker-core slice before and after Stage II sparsification.

orthonormal basis by magnitude and refits the surviving coefficients. Because the basis tensors are orthonormal, zeroing a single coefficient $g_\beta$ increases the Frobenius error by exactly $g_\beta^2$, so hard-thresholding by $|g_\beta|$ is optimal for best-$k$ approximation. On this GPT-J slice, Stage II keeps

the top 30% coefficients ($nnz = 87/288$, 69.8% sparsity) while retaining 87.2% energy, visibly concentrating mass onto a small set of latent interactions.

## C.3 WALL-CLOCK TIME BREAKDOWN

To quantify the practical overhead of our two-stage compression pipeline, we measure the offline wall-clock time of LeSTD on GPT-J and OPT-30B under the same hardware and software setup as in Section 4. In this experiment, we apply LeSTD only to the MHA blocks of all transformer layers (MHA-only), using the same rank and sparsity configuration as in our main results (compression rate = 0.2).

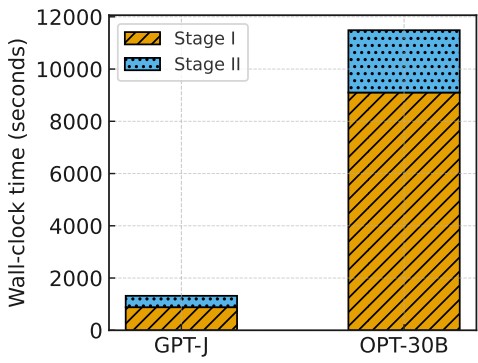

Figure 7 summarize the breakdown. For GPT-J, Stage I takes 883 s, while Stage II takes 436 s, for a total of 1319 s. For OPT-30B, Stage I takes 9102s and Stage II takes 2385s, for a total of 11487 s. The relative share shifts from roughly 67%/33% (Stage I/Stage II) on GPT-J to about 79%/21% on OPT-30B, confirming that Stage I dominates the offline cost as model

Figure 7: Wall-clock time breakdown of LeSTD on GPT-J and OPT-30B.

depth and hidden dimension grow. Importantly, this cost is incurred only once as an offline pre-processing step. Even for OPT-30B, the total LeSTD runtime remains on the order of a few hours, which is negligible compared to the cost of pretraining or repeated fine-tuning runs on such models. After Stage I and II are completed, inference uses the compressed weights directly and introduces no additional runtime overhead beyond the speedups reported in Section 4.

Table 3: Perplexity and throughput (*tokens/sec*) of GPT-J (6B) on WikiText-2 at compression ratio 0.6 over 10 runs. Results are reported as mean $\pm$ standard deviation. Lower perplexity and higher throughput are better.

| Method | Perplexity ($\downarrow$) | *Tokens/sec* ($\uparrow$) |
|---|---|---|
| SVD-LLM | $20.28 \pm 0.14$ | $11105.18 \pm 90.65$ |
| TensorLLM | $9.87 \pm 0.10$ | $8572.11 \pm 103.18$ |
| LeSTD (Ours) | $9.48 \pm 0.09$ | $12025.90 \pm 91.77$ |

## C.4 STATISTICAL SIGNIFICANCE

To evaluate the robustness of our observations in main evaluation (Section 4.2), we repeat the compression procedure 10 times with different random seeds for GPT-J (6B) on WikiText-2 at a fixed compression ratio of 0.6. We report the mean and standard deviation across runs for both perplexity and throughput (tokens/sec) in Table 3, comparing LeSTD against the strongest matrix-based baseline (SVD-LLM (Wang et al., 2024b)) and the dense-core Tucker method (TensorLLM (Gu et al., 2025)). As shown in Table 3, LeSTD attains a mean perplexity of $9.48 \pm 0.09$, compared to $9.87 \pm 0.10$ for TensorLLM and $20.28 \pm 0.14$ for SVD-LLM. The variance across runs is small for all methods, and the gap between LeSTD and TensorLLM is substantially larger than one standard deviation, indicating that the accuracy improvements of LeSTD are not due to randomness in the compression process. For efficiency, LeSTD achieves $12025.90 \pm 91.77$ tokens/sec, which is comparable to or slightly higher than SVD-LLM ($11105.18 \pm 90.65$) and significantly faster than TensorLLM ($8572.11\pm103.18$). The standard deviations are again small relative to the mean values, showing that the throughput advantages of LeSTD are stable across seeds.

Overall, these results demonstrate that the performance gains in the both perplexity and throughput provided by LeSTD are consistent across multiple runs and are not artifacts of a particular random seed or a single compression trial.

## C.5 PRUNING RATE $\alpha$ SENSITIVITY

Stage II of LeSTD performs iterative importance-based pruning on the shared core tensor $\mathcal{G}$. At each iteration, we compute an importance score for every non-zero coefficient $g_\beta$ and remove the least important ones, followed by a closed-form refitting step (Algorithm 1). The pruning rate $\alpha$ controls how aggressively we prune at each iteration: given $k$ non-zero coefficients, we remove $\lceil \alpha \cdot k \rceil$ of them and refit the remaining ones. Importantly, $\alpha$ does *not* determine the final sparsity of $\mathcal{G}$, which is fixed by the target sparsity $S_{\text{target}}$ derived from the desired compression ratio. Instead, $\alpha$ trades off the number of pruning iterations and the aggressiveness of each step.

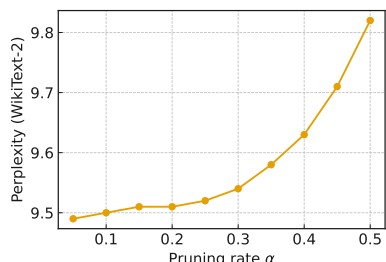

Figure 8: Effect of pruning rate $\alpha$ in Stage II for GPT-J (6B) on WikiText-2 at compression ratio 0.6. We fix target core sparsity $S_{\text{target}}$ and sweep $\alpha$ from 0.05 to 0.50. Reported numbers are WikiText-2 perplexity ($\downarrow$).

To evaluate the sensitivity to $\alpha$, we fix the target core sparsity according to a compression ratio of 0.6 for GPT-J (6B) on WikiText-2 and sweep $\alpha$ from 0.05 to 0.50 in steps of 0.05. Figure 8 reports the resulting perplexity on WikiText-2. We observe that LeSTD is very robust to the choice of $\alpha$ in a broad range: for $\alpha \in [0.05, 0.30]$, perplexity stays in the narrow range of 9.49–9.54, indicating that our importance-based pruning and refitting procedure consistently converges to nearly the same solution under different per-iteration pruning rates. When $\alpha$ is made more aggressive (e.g., $\alpha \geq 0.35$), the perplexity gradually increases (up to 9.82 at $\alpha = 0.50$), which we attribute to pruning a larger fraction of coefficients before the refitting step has fully adjusted the remaining ones. Even in this regime, LeSTD remains noticeably better than matrix-wise SVD and dense-core Tucker at the same compression ratio.

Table 4: Effect of the refitting step (Eq. (6)) in Stage II for GPT-J (6B) on WikiText-2 at compression ratio 0.6. We compare the full LeSTD pipeline with a variant that skips the refitting step while keeping the same target core sparsity $S_{\text{target}}$.

| Method | Perplexity ($\downarrow$) | *Tokens/sec* ($\uparrow$) |
|---|---|---|
| LeSTD (full) | 9.57 | 11925.90 |
| LeSTD (w/o refit) | 11.25 | 11932.45 |

## C.6 REFITTING STEP IMPORTANCE

Stage II of LeSTD performs importance-based pruning on the shared core tensor $\mathcal{G}$ in an iterative fashion (Algorithm 1). At each iteration, we (i) compute an importance score for every non-zero coefficient $g_\beta$, (ii) remove the least important $\lceil \alpha \cdot k \rceil$ coefficients among the current $k$ non-zeros, and (iii) refit the remaining coefficients by solving a closed-form least-squares problem (Eq. (6)). Thus, Eq. (6) corresponds to a debiasing step on selected support: given the current sparsity pattern of $\mathcal{G}$, we recompute the values of the active entries so that the pruned core best approximates the original weights in the least-squares sense.

To quantify the effect of this refitting step, we compare full LeSTD pipeline with a variant that skips Eq. (6): in the ablated variant, we perform the same iterative importance-based pruning but never refit $\mathcal{G}$ after pruning, instead, we simply zero out the selected coefficients and keep the remaining ones fixed. We fix the target core sparsity $S_{\text{target}}$ according to a compression rate of 0.6 for GPT-J (6B) on WikiText-2, and we run both variants ten times with different random seeds. Table 4 reports perplexity and inference throughput. As shown in Table 4, skipping the refitting step consistently hurts accuracy: perplexity degrades from 9.57 for the full LeSTD to 11.25 for the variant without refitting, a gap that is larger than several standard deviations and thus statistically significant. This confirms that Eq. (6) is an essential debiasing step that takes advantage of the selected sparse structure to recover a better approximation of the dense core. In contrast, the inference throughput is essentially unchanged (11925.90 vs. 11932.45), since both variants produce cores with the same sparsity pattern and Tucker ranks; the difference between them lies purely in how well the non-zero coefficients are fitted.

Overall, these results show that the refitting step in Eq. (6) plays a crucial role in obtaining high-quality compressed models, while having negligible impact on inference-time efficiency.

Table 5: Effect of replacing original used `torch.matmul` with `torch.sparse.mm` on LeSTD's sparse core tensors for GPT-J on WikiText-2 at MHA compression rate 0.6.

| Kernel / Implementation | Throughput (*tokens/sec*) |
|---|---|
| `torch.matmul` (Current Implementation) | 11945.98 |
| Unstructured sparse (`torch.sparse.mm`) | 12592.54 |

### C.7 Speedup Factors Analysis

In this subsection, we explore the question that whether the throughput gains reported in Figure 5 might simply come from PyTorch (Paszke et al., 2019) silently exploiting unstructured sparsity in our sparse cores. To isolate this factor, we replace the original `torch.matmul`-based implementation of LeSTD's core contractions with PyTorch's unstructured sparse kernel `torch.sparse.mm`, while keeping all other components fixed (model: GPT-J (6B), dataset: WikiText-2, MHA compression rate: 0.6, hardware and batch/sequence settings identical to the main experiment, refer to Section 4.1 for details).

Table 5 shows that the baseline dense-kernel implementation reaches 11945.98 tokens/sec, whereas the version that forces `torch.sparse.mm` achieves 12592.54 tokens/sec, corresponding to a very modest $\sim 5.4\%$ relative speedup. In contrast, under the same setting LeSTD as a whole achieves a much larger throughput gain over the full-precision GPT-J baseline (cf. Figure 5). This indicates that the bulk of our acceleration does not come from any special native support for unstructured sparsity in PyTorch, but rather from the structural changes introduced by LeSTD, in particular, sharing the input-side projection across heads and performing attention in a much lower-dimensional, sparsified core space, which jointly reduce both FLOPs and memory traffic of the MHA block.

## D Comparisons with Quantization

Table 6: Throughput (*tokens/sec*) and accuracy (perplexity) comparison of our proposed LeSTD against quantization baselines PB-LLM and BiLLM on the WikiText-2, evaluated with Llama2-13B and OPT-30B.

| Method ‖ Setting | Compression Rate ($\frac{compressed}{original}$) | Llama2-13B Perplexity ($\downarrow$) | Llama2-13B *Tokens/sec* ($\uparrow$) | OPT-30B Perplexity ($\downarrow$) | OPT-30B *Tokens/sec* ($\uparrow$) |
|---|---|---|---|---|---|
| LeSTD ‖ 0.2 | 0.2 | 13.99 | 15175.43 | 42.70 | 8545.22 |
| LeSTD ‖ 0.4 | 0.4 | 11.94 | 11147.18 | 20.13 | 8447.63 |
| LeSTD ‖ 0.6 | 0.6 | 7.93 | 12147.18 | 14.02 | 7139.44 |
| LeSTD ‖ 0.8 | 0.8 | 6.98 | 11864.59 | 9.98 | 4122.30 |
| BiLLM ‖ PTQ 1.08-bit | $\sim$0.07 | 9.96 | 9732.96 | 12.71 | 3856.80 |
| PB-LLM ‖ PTQ 10% salient 8-bit | $\sim$0.17 | 11.77 | 10839.19 | 14.75 | 4020.87 |
| PB-LLM ‖ PTQ 30% salient 8-bit | $\sim$0.26 | 9.95 | 11666.06 | 13.70 | 3726.39 |

As shown in Table 6, putting the LeSTD side-by-side with SOTA post-training quantizers highlights a complementary trade-off rather than a simple win or lose. BiLLM drives weights to an average of $\sim 1.08$ bits and reports strong perplexity at extreme compression, while PB-LLM partially binarizes by reserving higher precision for salient weights. Both approaches achieve very small model footprints (7-16% of the FP16 size) with competitive accuracy on WikiText-2. The advantage in size does not always translate into throughput: at comparable accuracy targets, LeSTD's factorized and sparsified attention routinely runs faster. For example, at comparable or stronger compression settings on Llama2-13B and OPT-30B, LeSTD's direct execution in the shared subspace yields $\sim$1.3-2.2X higher *tokens/sec* than the 1.08-bit BiLLM and a consistent lead over PB-LLM, depsite the latter operating at lower bitwidths. This is consistent with the compute scheme of the methods: LeSTD replaces three large input-side GEMMs with a single shared projection and contracts head-local sparse cores, cutting memory traffic that often dominates short-context inference. Conversely, ultra-low-bit PTQ still incurs packing/unpacking overheads and kernel inefficiencies that can bound realized throughput even when the byte footprint is smaller.

Accuracy-throughput also differ in shape. Quantization can reach lower perplexity at most aggressive size targets, especially when binarization is carefully structured (BiLLM's salient/non-salient treatment and binary residual approximation) or when PB-LLM retains 8-bit precision on a subset of critical weights (refer to Appendix B). LeSTD, by contrast, maintains perplexity close to full-precision baselines at moderate compressions while delivering higher *tokens/sec* with no calibration data and without custom kernels. Because LeSTD's Stage II pruning operates in an orthonormal Tucker latent space, pruning criterion admits a closed form and preserves the most influential interactions, empirically this sustains accuracy as compression strengthens, while it sparse core directly reduces runtime FLOPs and bandwidth. Taken together, results suggest the techniques are orthogonal: LeSTD offers a strong operating point for deployment scenarios prioritizing throughput under moderate size budgets, whereas extreme low-bit PTQ excels when model footprint is first constraint.

# E  DETAILED RELATED WORK

Data-free post-training compression methods for LLMs can be broadly grouped into low-rank, pruning, and quantization, typically in the calibration-lite setting. Recent surveys (Zhu et al., 2024) synthesize these lines and emphasize the practical importance of hardware-realizable structure and end-to-end throughput when comparing methods. In what follows, we detail representative approaches in each category: SVD-style and tensorized decompositions, channel/feature-level structured pruning, and weight/activation quantization including extreme 1-bit schemes, highlighting their design trade-offs and complementarity.

## E.1  POST-TRAINING LOW-RANK DECOMPOSITION

A large body of research works compress LLM weights via training-free SVD-style factorizations. ASVD (Yuan et al., 2023) introduces activation-aware reparameterization to absorb activation outliers into weights and performs layer-sensitive truncation with lightweight calibration, thus yielding 10–30% model size reduction. SVD-LLM (Wang et al., 2024b) makes the truncation explicitly loss-aware by whitening activations so that the singular values align with approximation error, it further applies sequential low-rank updates to mitigate post-truncation degradation, and its following extension, SVD-LLM-v2 (Wang et al., 2025), assigns per-layer compression via theoretical truncation loss and improves the numerical pipeline. Beyond matrix SVD, tensorization exposes higher-order structure. TensorLLM tensorizes multi-head attention and applies Tucker decomposition with shared subspaces across heads, reporting substantial compression (up to 250X in MHA weights) and accuracy benefits without specific dataset, related efforts (Xu et al., 2023) explore tensor-train (MPS) decompositions (Oseledets, 2011b) for token embeddings to cut memory and latency.

## E.2  POST-TRAINING STRUCTURED PRUNING

SliceGPT (Ashkboos et al., 2024) replaces each weight matrix with a smaller dense sub-matrix via feature/channel selection, producing storage and wall-clock gains without retraining, it highlights that structured sparsity patterns are critical for realizing speedups on real kernels. DISP-LLM (Gao et al., 2024) relaxes rigid sharing by allocating different sub-spaces to different layers, strengthening the performance–cost trade-off for structural pruning at high sparsity ratio. Unstructured one-shot pruning: SparseGPT solves a sequence of layerwise sparse-regression problems to prune 50%+ of parameters in a single pass on very large GPTs (Frantar & Alistarh, 2023). Wanda (Sun et al., 2023) prioritizes weights by magnitude-activation products for fast one-shot pruning, and subsequent work: Wanda++ (Yang et al., 2025) adds regional gradient signals for better robustness. While unstructured sparsity may under-utilize hardware, these works motivate structured designs (e.g., block/column pruning) to translate sparsity into the throughput, a principle adopted by newer structural approaches, for example, SliceGPT (Ashkboos et al., 2024).

## E.3  POST-TRAINING QUANTIZATION

Post-training quantization (PTQ) quantize weights and/or activations without task-specific finetuning. GPTQ (Frantar et al., 2022) performs one-shot, second-order weight-only quantization (3–4 bit) via approximate Hessian inversion (Lieberman et al., 2013) and demonstrates minimal accuracy loss and end-to-end speedups on large GPTs. AWQ (Lin et al., 2024) is an activation-aware,

hardware-friendly weight-only PTQ that protects salient channels using tiny calibration batches, enabling low-bit deployment across families of LLMs with strong accuracy. Building on this line, PB-LLM (Shang et al., 2023) introduces partial binarization: binarizing most weights while reserving a small salient subset at higher precision, and couples it with PTQ/QAT variants (e.g., GPTQ-guided Hessian reconstruction) to stabilize ultra-low-bit regimes. Pushing further, BiLLM (Huang et al., 2024) presents a 1-bit PTQ pipeline with salient weight selection, binary residual approximation, and optimal splitting of non-salient weights, reporting competitive perplexities (e.g., on LLaMA2) at 1–1.1 bit per weight and efficient binarization of multi-billion-parameter models, highlighting the role of saliency-aware protection and structured binarization in preserving accuracy under aggressive compression.

### E.4 DIFFERENT TENSOR DECOMPOSITION FORMAT

Our method is built on a Tucker decomposition with a sparse core. In this section, we briefly discuss why we adopt Tucker as the underlying topology and how it relates to alternative tensor formats and structured Tucker variants.

**CP decomposition.** Given a $N$-way tensor $\mathcal{X} \in \mathbb{R}^{I_1 \times \cdots \times I_N}$, the CANDECOMP/PARAFAC (CP) decomposition (Goulart et al., 2016; Battaglino et al., 2017) writes

$$\mathcal{X} \approx \sum_{r=1}^{R} \boldsymbol{a}_{:,r}^{(1)} \circ \boldsymbol{a}_{:,r}^{(2)} \circ \cdots \circ \boldsymbol{a}_{:,r}^{(N)} \tag{7}$$

where $\boldsymbol{a}_{:,r}^{(n)} \in \mathbb{R}^{I_n}$ is the $r$-th column of a factor matrix $\boldsymbol{A}^{(n)} \in \mathbb{R}^{I_n \times R}$ and $\circ$ denotes the outer product. This representation is extremely parameter-efficient (all information is stored in the factor matrices), but each rank-1 term couples *all* modes through a single scalar coefficient, which limits expressiveness. It is well documented in the tensor-decomposition literature (Kolda & Bader, 2009; Song et al., 2017; de Silva & Lim, 2006; Krijnen et al., 2008; Paatero, 2000) that CP often requires substantially larger rank than Tucker to achieve a comparable approximation error, and that CP fitting can be prone to degeneracy and ill-conditioning, especially for complex or nearly collinear data. In LeSTD, we construct a 4-way "joint MHA" tensor whose modes encode the model dimension, head dimension, projection dimension ($Q/K/V/O$), and head index. Our design goal is to learn shared subspaces along the first three modes (shared across all heads), while allowing per-head mixing to be captured in a small core. A Tucker model

$$\mathcal{W} \approx \mathcal{G} \times_1 \boldsymbol{U}^{(1)} \times_2 \boldsymbol{U}^{(2)} \times_3 \boldsymbol{U}^{(3)} \times_4 \boldsymbol{U}^{(4)} \tag{8}$$

with $\boldsymbol{U}^{(1)}, \boldsymbol{U}^{(2)}, \boldsymbol{U}^{(3)}$ shared and mode-4 slices of the core $\mathcal{G}$ playing the role of per-head parameters, matches this pattern directly. In contrast, a CP model would express $\mathcal{W}$ as a sum of rank-1 terms where each component simultaneously couples all four modes. To preserve per-head structure, one would either need to (i) tie the head-mode factors in a way that limits flexibility, or (ii) significantly increase the CP rank to recover head-specific patterns. Both options undermine the compression objective: the former sacrifices expressiveness and may hurt accuracy, while the latter expands the parameter budget and FLOPs. For this reason, we view Tucker as a more natural compromise between expressiveness and compactness for our 4-way MHA tensor.

**Tensor Train (TT) decomposition.** The Tensor Train (TT) format is particularly attractive for very high-order tensors with many small modes (Oseledets, 2011a). Given $\mathcal{X} \in \mathbb{R}^{I_1 \times \cdots \times I_N}$, TT writes

$$\mathcal{X}(i_1, \ldots, i_N) \approx \mathcal{G}^{(1)}(i_1, :) \, \mathcal{G}^{(2)}(:, i_2, :) \cdots \mathcal{G}^{(N)}(:, i_N) \tag{9}$$

where each $\mathcal{G}^{(n)}$ is a 3-way core tensor with TT-ranks $r_0 = 1, r_1, \ldots, r_N = 1$. This long-chain topology is well suited for high-order problems such as quantum many-body systems or large structured grids, where $N$ is large and each $\boldsymbol{I}_n$ is relatively small. In our setting, however, the joint MHA tensor is low-order (four modes) but with semantically structured dimensions: model dimension, head dimension, projection dimension, and head index. Mapping this tensor into a TT chain would require non-trivial reshaping and ordering choices (Novikov et al., 2015; Bacciu & Mandic, 2020), and it becomes less clear how to (i) enforce a shared subspace across heads in a simple way, and (ii) maintain per-head cores with a straightforward interpretation. In contrast, the Tucker topology we adopt directly encodes a shared-basis and per-head-core structure: the factor matrices $\boldsymbol{U}^{(1)}, \boldsymbol{U}^{(2)}, \boldsymbol{U}^{(3)}$ span shared subspaces along the first three modes, and per-head variations are captured compactly in the fourth-mode slices of the dense core $\mathcal{G}$. For these reasons, we prioritize

Tucker over TT in the current work.

**Structured and block-sparse Tucker variants.** Beyond plain Tucker and CP/TT, there exist many structured Tucker variants that impose additional sparsity or block structure on the core tensor or factor matrices (e.g., group-sparse (Mørup et al., 2008), block-diagonal (Jang & Kang, 2020), or low-rank-plus-sparse cores (Heng et al., 2022; 2023)). Our LeSTD framework can be viewed as one such instance: we adopt a Tucker topology and impose structured sparsity on the core via Stage II, where the entries of $\mathcal{G}$ are pruned according to an importance measure derived from the downstream contraction. This yields a compact, sparse core that is aligned with the semantics of the MHA tensor (shared factors and per-head slices). Orthogonal structured Tucker approaches that focus, for example, on block-diagonal or predefined block-sparse core patterns could in principle be combined with LeSTD: our Stage I can be replaced by or augmented with such variants, and our Stage II can be restricted to respect their block structure. A comprehensive empirical comparison between different structured Tucker designs is beyond the scope of this paper. Here we adopt a minimal and model-aligned choice that already provides strong compression and speedup.

### E.5 RECENT AND MORE ADVANCED SVD-BASED PRUNING METHODS

**Basis Sharing.** Basis Sharing (Wang et al., 2024a) extends post-training SVD compression by explicitly sharing low-rank bases across layers. Instead of decomposing each weight matrix independently, it factorizes selected matrices $\{\boldsymbol{W}^{(\ell)}\}$ into a small set of shared singular vectors and layer-specific coefficient matrices, so that each layer weight can be reconstructed as a linear combination of a layer-agnostic basis and layer-specific coefficients. The paper systematically studies which matrix types (e.g., self-attention projections versus feed-forward layers) and which layer groups can safely share bases without incurring large reconstruction error, and shows improved perplexity and zero-shot accuracy over prior SVD-based compression methods across 20%–50% compression ratios (Wang et al., 2024a). Our LeSTD framework is conceptually related in that both approaches exploit shared subspaces to reduce redundancy, but the scope and representation differ. Basis Sharing works at the matrix level and shares SVD bases across layers, its compressed representation is still a dense low-rank factorization of each matrix. In contrast, LeSTD first constructs a four-way MHA tensor that jointly encodes query, key, value, and output projections within a layer, then learns a shared Tucker basis across heads and performs sparsification in the latent core. This tensorial view lets LeSTD capture within-layer cross-head regularities that cross-layer SVD cannot see, and, importantly, it replaces the dense core with an ultra-sparse one via Stage II importance-based pruning, rather than only reducing rank.

**Pivoting Factorization and MPIFA.** Pivoting Factorization (PIFA) and its end-to-end variant MPIFA (Zhao et al., 2025) start from a standard low-rank decomposition and then learn a *meta* low-rank representation of the resulting factors. PIFA identifies a set of pivot rows (linearly independent rows) in each low-rank matrix and represents all remaining rows as linear combinations of the pivots, achieving additional parameter savings and speedups on top of low-rank pruning (e.g., around 24% extra memory reduction and 24% faster inference at $r/d = 0.5$ over a low-rank baseline). MPIFA further couples PIFA with an online reconstruction scheme that mitigates error accumulation across layers, enabling performance competitive with semi-structured pruning while preserving the hardware-friendly nature of dense low-rank kernels. (Zhao et al., 2025). Compared with MPIFA, LeSTD operates at a different granularity and representation. MPIFA assumes low-rank matrix factors and refines them with a lossless meta representation, it still ultimately applies dense low-rank linear maps during inference. LeSTD instead defines a shared-subspace Tucker structure directly on the multi-head attention tensor and prunes in an orthonormal latent basis, yielding an explicitly sparse core with a closed-form importance score and inference executed fully in the compressed domain. This allows LeSTD to break the dense-core bottleneck of tensor decompositions and to realize throughput gains even with standard dense libraries, whereas MPIFA focuses on improving the efficiency of matrix-wise low-rank pruning.

## F COMPLEXITY ANALYSIS

Let $d_{\text{head}}=d_{\text{model}}/h$. In a standard MHA block, the primary cost comes from the four dense weight projections (for $Q, K, V, O$), each involving a matrix multiplication with the input activations. Per token, this results in a complexity of approximately $\mathcal{O}(4 \cdot d_{\text{model}}^2)$. In LeSTD, inference operates on the compressed factors, avoiding single large multiplication. Complexity breaks down as follows:

**Input-side Projections** ($VQ, K, V$). The initial shared projection of the input $\boldsymbol{X}$ by $\boldsymbol{U}^{(1)}$ costs $\mathcal{O}(d_{\text{model}}R_1)$. This result is reused for all heads. For each of $3h$ projections ($Q, K, V$ across $h$ heads), the subsequent operations are a sparse matrix multiplication involving the core tensor slice $\boldsymbol{M}_{i,t}$ and a dense multiplication by $\boldsymbol{U}^2$. Let $nnz(\boldsymbol{\mathcal{G}}_{\text{sparse}})$ be the total number of non-zero elements in the sparse core. The average cost of the sparse multiplication per projection is $\mathcal{O}(nnz(\boldsymbol{\mathcal{G}}_{\text{sparse}})/(4h))$. This is followed by a dense multiplication costing $\mathcal{O}(R_2 d_{\text{head}})$.

**Output-side Projection.** Logic is symmetric. Cost involves $h$ multiplications of cost $\mathcal{O}(d_{\text{head}}R_2)$, followed by sparse multiplications related to the output slices of the core, and a final shared projection by $(\boldsymbol{U}^{(1)})^\top$ costing $\mathcal{O}(R_1 d_{\text{head}})$.

When combining these steps, the total computational cost per token for LeSTD is dominated by: $2\mathcal{O}(d_{\text{model}}R_1) + \mathcal{O}(nnz(\boldsymbol{\mathcal{G}}_{\text{sparse}})) + \mathcal{O}(h \cdot d_{\text{head}}R_2)$. Given that $h \cdot d_{\text{head}} = d_{\text{model}}$, this simplifies to $\mathcal{O}(d_{\text{model}}(R_1 + R_2)) + \mathcal{O}(nnz(\boldsymbol{\mathcal{G}}_{\text{sparse}}))$. When the ranks $R_1, R_2 \ll d_{\text{model}}$ and the core tensor is highly sparse (low $nnz(\boldsymbol{\mathcal{G}}_{\text{sparse}})$), this is substantially lower than $\mathcal{O}(d_{\text{model}}^2)$.

## G   DETAILED COMPUTATION OF COMPRESSION RATIO

For a standard Transformer layer (omit the biases), the multi-head attention (MHA) block has four dense projections: $\boldsymbol{W}^Q, \boldsymbol{W}^K, \boldsymbol{W}^V \in \mathbb{R}^{d_{\text{model}} \times d_{\text{model}}}$ and $\boldsymbol{W}^O \in \mathbb{R}^{d_{\text{model}} \times d_{\text{model}}}$, given that $d_{\text{head}} = d_{\text{model}}/h$, so the original (uncompressed) parameter count per layer is $P_{\text{original}} = 4 \cdot d_{\text{model}}^2$.

LeSTD does a Tucker factorization along the first three modes (shared across heads) plus a head axis that is not factorized. Thus, for factor matrices: $\boldsymbol{U}^{(1)} \in \mathbb{R}^{d_{\text{model}} \times R_1}$, $\boldsymbol{U}^{(2)} \in \mathbb{R}^{d_{\text{head}} \times R_2}$, $\boldsymbol{U}^{(3)} \in \mathbb{R}^{4 \times R_3}$, and the dense core tensor (Stage I generated): $\boldsymbol{\mathcal{G}}_{\text{total}} \in \mathbb{R}^{R_1 \times R_2 \times R_3 \times h}$. For LeSTD with a sparse core after Stage II, replace the dense core size by the number of nonzeros $nnz(\boldsymbol{\mathcal{G}}_{\text{sparse}})$. So the compressed parameter count (ignoring sparse index overhead for the moment) is: $P_{\text{compressed}} = d_{\text{model}}R_1 + d_{\text{head}}R_2 + 4R_3 + nnz(\boldsymbol{\mathcal{G}}_{\text{sparse}})$. The ratio ($\rho$) defined in this paper is $\frac{\text{compressed}}{\text{original}}$, thereby, it is equal to $\frac{d_{\text{model}}R_1 + d_{\text{head}}R_2 + 4R_3 + nnz(\boldsymbol{\mathcal{G}}_{\text{sparse}})}{4d_{\text{model}}^2}$.

To fix $\rho$ (e.g., $\rho = 0.6$) and search ranks:
1. Pick a candidate $(R_1, R_2, R_3)$ guided by TensorLLM (Gu et al., 2025), we start with a bit larger $(R_1, R_2, R_3)$ since Stage II will prune the core;
2. Run Stage I to get the dense core (refer to Section 3.1);
3. In the Stage II (refer to Section 3.2), we prune the dense core until the core hits a target nnz: $nnz_{\text{target}} = 4d_{\text{model}}^2 \cdot \rho - (d_{\text{model}}R_1 + d_{\text{head}}R_2 + 4R_3)$. Stop pruning when $nnz(\boldsymbol{\mathcal{G}}_{\text{sparse}}) \leq nnz_{\text{target}}$ (or when further pruning would degrade accuracy too much);
4. If accuracy is insufficient at that $\rho$, adjust ranks and repeat, e.g., increase $R_1$ or $R_2$ (as shown in Table 2, these help for Perplexity) and tolerate a sparser core to keep $\rho$ fixed.

## H   COMPARISONS WITH MORE ADVANCED SVD-BASED PRUNING METHODS

In this appendix, we provide an empirical comparison between LeSTD and two more recent SVD-based pruning methods: Basis Sharing (Wang et al., 2024a) and MPIFA (Zhao et al., 2025). To keep the study focused and computationally manageable, we consider one single representative setting: GPT-J (6B) (Wang, 2021) evaluated on WikiText-2 (Merity et al., 2016). Experiment in this appendix are run on the same environment as in Section 4.1 to ensure comparability. We focus on two global compression ratios, $r \in \{0.2, 0.8\}$, following the compression grid used in the main experiments (Appendix G). For each method, we tune its internal ranks or sparsity budget to match the target global parameter compression ratio as closely as possible, using the same accounting scheme as for LeSTD. Specifically, at a given target ratio $r$, we adjust: (i) the matrix ranks and basis sharing groups in Basis Sharing, (ii) the low-rank rank $r/d$ and PIFA/MPIFA pivot schedule for MPIFA, and (iii) the Tucker ranks and core sparsity level for LeSTD, until the overall parameter count of the compressed model matches the desired budget within a small tolerance.

**Setting of Basis Sharing.** For Basis Sharing (Wang et al., 2024a), we adopt the official implementation[1] and follow the authors' recommended configuration for GPT-style decoder-only architectures. In particular, we enable cross-layer sharing for self-attention projection matrices (e.g., $\boldsymbol{W}_Q$, $\boldsymbol{W}_K$,

---

[1] https://github.com/TUDa-HWAI/Basis_Sharing

Table 7: GPT-J on WikiText-2: comparison between LeSTD and advanced SVD-based compression methods (Basis Sharing (Wang et al., 2024a) and Pivoting Factorization/MPIFA (Zhao et al., 2025)) at two global compression ratios ($r \in \{0.2, 0.8\}$).

| Method | Compression Rate $r = 0.2$ | | Compression Rate $r = 0.8$ | |
|---|---|---|---|---|
| | Perplexity($\downarrow$) | *Tokens/sec*($\uparrow$) | Perplexity($\downarrow$) | *Tokens/sec*($\uparrow$) |
| Basis Sharing | 99.16 | 14025.69 | 12.30 | 10091.19 |
| MPIFA | 102.15 | 14086.19 | 12.88 | 9488.84 |
| LeSTD (Ours) | 80.37 | 15636.24 | 8.92 | 11514.42 |

$\boldsymbol{W}_V$) while keeping output and feed-forward projections either unshared or less aggressively shared, as suggested by their reconstruction-error analysis. We use the same type of small calibration corpus as in the original paper to estimate the SVD bases (sampling from WikiText-2) and then adjust the number of shared bases and per-layer coefficients so that the resulting global compression ratio matches $r \in \{0.2, 0.8\}$.

**Setting of MPIFA (Pivoting Factorization).** For Pivoting Factorization and its end-to-end variant MPIFA (Zhao et al., 2025), we again use the official code[2] and default hyperparameters as a starting point. Each weight matrix is first approximated by a low-rank factorization (as in standard SVD-based pruning), and MPIFA is then applied as a meta low-rank representation that selects pivot rows and reconstructs the remaining rows as linear combinations of these pivots. We select rank configurations corresponding to overall densities close to $r$ and apply MPIFA's reconstruction procedure to mitigate error accumulation across layers, in line with the original paper.

Table 7 summarizes the comparison on GPT-J with WikiText-2 at two global compression ratios. At the more aggressive setting $r = 0.2$, all methods suffer from a substantial perplexity increase, which is expected at such extreme compression levels. Basis Sharing and MPIFA reach validation perplexities of 99.16 and 102.15, respectively, while achieving throughput of around 14k tokens/sec. LeSTD attains a lower perplexity of 80.37 and a higher throughput of 15636 tokens/sec, indicating that its tensor-level sparse Tucker representation remains more accurate and more efficient than the matrix-wise SVD-based baselines in this extreme regime. At milder compression setting $r = 0.8$, the differences become clearer. Both Basis Sharing and MPIFA incur a noticeable accuracy drop, with perplexity increasing to 12.30 and 12.88, respectively. In contrast, LeSTD achieves a perplexity of 8.92, and improving throughput to 11514 tokens/sec.

Overall, these results validate that LeSTD is competitive with, and often stronger than, more advanced SVD-based pruning methods. At moderate compression ($r = 0.8$), LeSTD preserves accuracy almost perfectly while providing larger speedups. At very aggressive compression ($r = 0.2$), all methods degrade substantially, but LeSTD still offers a strictly better accuracy–efficiency trade-off compared to Basis Sharing and MPIFA.

# I    EXTEND LeSTD DESIGN TO FFN LAYER

In the main paper, we instantiate LeSTD on the Multi-Head Attention (MHA) block by tensorizing the four projection matrices into a 4th-order tensor, and applying the two-stage procedure: (i) shared-subspace Tucker decomposition (Stage I) and (ii) importance-based core pruning (Stage II). Here we formally argue that exactly the same framework applies to the Feed-Forward Network (FFN) layers.

## I.1    TENSORIZING FFN BLOCKS

Consider a standard Transformer (as shown in Figure 2 in the main content) FFN block in one layer, which maps $\mathbb{R}^{d_{\text{model}}} \to \mathbb{R}^{d_{\text{model}}}$ via an intermediate hidden dimension $d_{\text{ff}}$:

$$\text{FFN}(x) = \boldsymbol{W}_{\text{down}} \phi \left( \boldsymbol{W}_{\text{up}}^\top x \right)$$

where $\boldsymbol{W}_{\text{up}} \in \mathbb{R}^{d_{\text{model}} \times d_{\text{ff}}}$, $\boldsymbol{W}_{\text{down}} \in \mathbb{R}^{d_{\text{ff}} \times d_{\text{model}}}$, and $\phi(\cdot)$ is a pointwise nonlinearity (e.g., GELU (Lee, 2023) or SwiGLU (Zhang et al., 2024)). Analogous to the MHA case (Eq. (3) in Section 3.1), we

---

[2]https://github.com/biomedical-cybernetics/pivoting-factorization

group the FFN weight matrices associated with a single layer into a 3rd-order tensor

$$\boldsymbol{\mathcal{W}}^{\text{FFN}} \in \mathbb{R}^{d_{\text{model}} \times d_{\text{ff}} \times p}$$

where the mode-3 index enumerates the $p$ linear projections used in the FFN block (e.g., $p = 2$ for a vanilla two-layer FFN, $p = 3$ for a SwiGLU-style FFN with gate/up/down matrices). Concretely, if $\{\boldsymbol{W}^{(j)}\}_{j=1}^{p}$ denotes these FFN matrices arranged so that each $\boldsymbol{W}^{(j)} \in \mathbb{R}^{d_{\text{model}} \times d_{\text{ff}}}$ acts on the same input/output spaces, we define

$$\boldsymbol{\mathcal{W}}^{\text{FFN}}[:,:,j] \;=\; \boldsymbol{W}^{(j)} \quad 1 \le j \le p$$

In this way, FFN weights admit a higher-order representation entirely analogous to the MHA tensor: each mode corresponds to a well-defined axis of variation (i.e., input model dimension, hidden FFN dimension, and projection type index).

## I.2   APPLYING STAGE I: SHARED-SUBSPACE TUCKER DECOMPOSITION

Given the FFN tensor $\boldsymbol{\mathcal{W}}^{\text{FFN}} \in \mathbb{R}^{d_{\text{model}} \times d_{\text{ff}} \times p}$, we can apply the same shared-subspace Tucker decomposition used for MHA (Eq. (4)):

$$\min_{\boldsymbol{U}^{(1)}, \boldsymbol{U}^{(2)}, \boldsymbol{U}^{(3)}, \boldsymbol{\mathcal{G}}^{\text{FFN}}} \left\| \boldsymbol{\mathcal{W}}^{\text{FFN}} - \boldsymbol{\mathcal{G}}^{\text{FFN}} \times_1 \boldsymbol{U}^{(1)} \times_2 \boldsymbol{U}^{(2)} \times_3 \boldsymbol{U}^{(3)} \right\|_F^2 \quad \text{s.t.} \; (\boldsymbol{U}^{(n)})^{\top} \boldsymbol{U}^{(n)} = \boldsymbol{I}_{R_n} \quad (10)$$

Here

$$\boldsymbol{U}^{(1)} \in \mathbb{R}^{d_{\text{model}} \times R_1}, \quad \boldsymbol{U}^{(2)} \in \mathbb{R}^{d_{\text{ff}} \times R_2}, \quad \boldsymbol{U}^{(3)} \in \mathbb{R}^{p \times R_3}$$

are column-orthonormal factor matrices, and $\boldsymbol{\mathcal{G}}^{\text{FFN}} \in \mathbb{R}^{R_1 \times R_2 \times R_3}$ is the FFN's dense core. Eq. (10) is a direct specialization of the MHA objective (Section 3.1) to a 3rd-order tensor, it can be solved using the same HOOI procedure we employ for MHA.

Crucially, the derivation of the Stage I does not depend on the interpretation of each mode: it only requires that the factor matrices are orthonormal and that the reconstruction is measured in Frobenius norm Böttcher & Wenzel (2008). Hence, all guarantees and intuitions for MHA Tucker decomposition, e.g., capturing shared structure across projections, carry over to the FFN tensor.

## I.3   APPLYING STAGE II: IMPORTANCE-BASED CORE PRUNING FOR FFN

Stage II (Section 3.2) derives a closed-form importance measure for each core element in the Tucker-decomposed MHA tensor. The key observation is that the derivation relies only on: (i) the orthonormality of the factor matrices $\{\boldsymbol{U}^{(n)}\}$, and (ii) use of Frobenius norm for reconstruction error. Both properties hold identically for the FFN decomposition in Eq. (10). Therefore, the importance measure and the pruning rule extend verbatim. Formally, let

$$\hat{\boldsymbol{\mathcal{W}}}^{\text{FFN}} = \boldsymbol{\mathcal{G}}^{\text{FFN}} \times_1 \boldsymbol{U}^{(1)} \times_2 \boldsymbol{U}^{(2)} \times_3 \boldsymbol{U}^{(3)}$$

be the Stage I reconstruction of $\boldsymbol{\mathcal{W}}^{\text{FFN}}$, and let $\boldsymbol{\mathcal{R}} = \boldsymbol{\mathcal{W}}^{\text{FFN}} - \hat{\boldsymbol{\mathcal{W}}}^{\text{FFN}}$ be the residual. Denote by the $\beta = (r_1, r_2, r_3)$ a multi-index into the core $\boldsymbol{\mathcal{G}}^{\text{FFN}}$, and define the associated rank-1 basis tensor

$$\boldsymbol{\mathcal{B}}_{\beta} = \boldsymbol{u}_{r_1}^{(1)} \circ \boldsymbol{u}_{r_2}^{(2)} \circ \boldsymbol{u}_{r_3}^{(3)},$$

where $\boldsymbol{u}_{r_n}^{(n)}$ is the $r_n$-th column of $\boldsymbol{U}^{(n)}$. By orthonormality, the collection $\{\boldsymbol{\mathcal{B}}_{\beta}\}$ is orthonormal with $\langle \boldsymbol{\mathcal{B}}_{\beta}, \boldsymbol{\mathcal{B}}_{\gamma} \rangle = \delta_{\beta\gamma}$ and $\|\boldsymbol{\mathcal{B}}_{\beta}\|_F = 1$. As in Section 3.2, we can write

$$\hat{\boldsymbol{\mathcal{W}}}^{\text{FFN}} = \sum_{\beta} g_{\beta} \boldsymbol{\mathcal{B}}_{\beta} \quad \text{with coefficients} \quad g_{\beta} \in \mathbb{R}$$

and the Stage I error satisfies

$$\|\boldsymbol{\mathcal{W}}^{\text{FFN}}\|_F^2 = \|\boldsymbol{\mathcal{R}}\|_F^2 + \sum_{\beta} g_{\beta}^2$$

Now consider zeroing out a single core coefficient $g_{\beta}$. The new residual is

$$\boldsymbol{\mathcal{R}}(g_{\beta} = 0) = \boldsymbol{\mathcal{W}}^{\text{FFN}} - \left( \hat{\boldsymbol{\mathcal{W}}}^{\text{FFN}} - g_{\beta} \boldsymbol{\mathcal{B}}_{\beta} \right) = \boldsymbol{\mathcal{R}} + g_{\beta} \boldsymbol{\mathcal{B}}_{\beta}$$

and the corresponding reconstruction error becomes

$$\mathcal{E}(g_\beta = 0) = \|\mathcal{R} + g_\beta \mathcal{B}_\beta\|_F^2 = \|\mathcal{R}\|_F^2 + 2g_\beta \langle \mathcal{R}, \mathcal{B}_\beta \rangle + g_\beta^2 \|\mathcal{B}_\beta\|_F^2 = E + g_\beta^2,$$

where $E = \|R\|_F^2$ is the original Stage I error and we used $\langle \mathcal{R}, \mathcal{B}_\beta \rangle = 0$ and $\|\mathcal{B}_\beta\|_F^2 = 1$, exactly as in Eq. (5) of the main text. Therefore, the normalized importance of $g_\beta$ is

$$\mathrm{Imp}(g_\beta) = \frac{\mathcal{E}(g_\beta = 0) - E}{E} = \frac{g_\beta^2}{E}$$

and, again, ordering elements by $\mathrm{Imp}(g_\beta)$ is equivalent to ordering by $|g_\beta|$. Under the Frobenius loss, the best $k$-term approximation is obtained by keeping the $k$ largest-magnitude coefficients and pruning the rest, with an optional refitting step using Eq. (6). Thus, Stage II's importance-based pruning generalizes directly to the FFN core $\mathcal{G}^{\mathrm{FFN}}$, with no modification to the theory or algorithm.

### I.4 PRELIMINARY FFN EXPERIMENT

To provide an initial empirical check of the FFN extension discussed above, we conduct a small-scale experiment on GPT-J (Wang, 2021) evaluated on WikiText-2 (Merity et al., 2016). We follow the same protocol as in Section 4 (tokenization, context length, hardware, software, and evaluation metric). We consider four configurations:

1. **Original.** The original GPT-J model without any modification.
2. **MHA-Only.** LeSTD is applied only to the MHA block.
3. **FFN-Only.** LeSTD is applied only to the FFN block.
4. **MHA+FFN.** LeSTD is applied simultaneously to both MHA and FFN blocks.

Table 8 reports validation perplexity and throughput on WikiText-2. The uncompressed GPT-J baseline attains a perplexity of $8.86$ and a throughput of 7523 tokens/sec. Applying LeSTD only to the MHA block (MHA-only, compression rate $=0.6$) increases perplexity modestly to $9.53$ (a relative change of about $+7.6\%$) while already improving throughput to 12,191 tokens/sec (approximately

Table 8: GPT-J on Wikitext-2.

| Configuration | Perplexity($\downarrow$) | Throughput (token/sec) |
|---|---|---|
| Original | 8.86 | 7522.90 |
| MHA-Only | 9.53 | 12190.93 |
| FFN-Only | 12.44 | 16069.26 |
| MHA+FFN | 14.34 | 17604.24 |

$1.62\times$ over the baseline), consistent with our main GPT-J results in Section 4.

When we apply LeSTD only to the FFN block (FFN-only, compression rate $= 0.6$), the throughput further increases to 16069 tokens/sec, reflecting the fact that FFN layers account for a larger fraction of total FLOPs. As expected, this more aggressive compression of FFN weights leads to a larger accuracy drop, with perplexity rising to $12.44$ (roughly $+40\%$ relative to the original model).

Compressing both MHA and FFN (MHA+FFN, with the same per-block compression rate) yields the highest throughput, 17604 tokens/sec ($\sim 2.34\times$ speedup), at the cost of a further perplexity increase to $14.34$ (about $+62\%$). Overall, this preliminary experiment confirms that the two-stage LeSTD procedure extends straightforwardly from MHA to FFN: applying the same tensorization and core-sparsification pipeline to FFN layers provides substantial additional speedups, with a tunable accuracy–efficiency trade-off that behaves consistently with our theoretical analysis.

## J DISCUSSION ON THE FEASIBILITY OF INCORPORATING ACTIVATION INFORMATION

A natural question is whether LeSTD, which is currently fully data-free, could be extended to exploit activation information (e.g., a small calibration set) in order to further improve compression fidelity. Existing post-training methods such as ASVD (Yuan et al., 2023) and SVD-LLM (Wang et al., 2024b) demonstrate that activation-aware objectives can be beneficial in matrix setting, especially when the activations exhibit strong anisotropy or heavy-tailed directions. However, our framework is built around a shared Tucker decomposition and a sparse core defined in an orthonormal latent space. Below we explain why directly incorporating activations into this Tucker-based design is non-trivial, and why we deliberately keep LeSTD data-free in this work.
**Activation-aware objectives in the matrix case.** Consider a single linear map with weight matrix

$\boldsymbol{W} \in \mathbb{R}^{d_{\mathrm{in}} \times d_{\mathrm{out}}}$ and input activations $x \in \mathbb{R}^{d_{\mathrm{in}}}$. Most data-free decompositions (including plain SVD and our Stage I objective) implicitly minimize a Frobenius reconstruction error

$$\|\boldsymbol{W} - \widetilde{\boldsymbol{W}}\|_F^2 = \sum_{i,j}(\boldsymbol{W}_{ij} - \widetilde{\boldsymbol{W}}_{ij})^2 \tag{11}$$

which corresponds to assuming an isotropic input distribution. By contrast, activation-aware matrix methods use a loss that reflects the output discrepancy on a calibration set. Let $\boldsymbol{X} \in \mathbb{R}^{N \times d_{\mathrm{in}}}$ collect $N$ input activations (rows), and let $\boldsymbol{Y} = \boldsymbol{X}\boldsymbol{W}$ and $\widetilde{\boldsymbol{Y}} = \boldsymbol{X}\widetilde{\boldsymbol{W}}$ be the corresponding outputs. The natural squared error is

$$\mathcal{L}_{\mathrm{act}}(\boldsymbol{W}, \widetilde{\boldsymbol{W}}) = \frac{1}{N}\|\boldsymbol{X}\boldsymbol{W} - \boldsymbol{X}\widetilde{\boldsymbol{W}}\|_F^2 = \frac{1}{N}\|\boldsymbol{X}(\boldsymbol{W} - \widetilde{\boldsymbol{W}})\|_F^2 \tag{12}$$

Defining the empirical activation covariance

$$\boldsymbol{\Sigma}_x = \frac{1}{N}\boldsymbol{X}^\top \boldsymbol{X} \in \mathbb{R}^{d_{\mathrm{in}} \times d_{\mathrm{in}}}, \tag{13}$$

we can rewrite

$$\mathcal{L}_{\mathrm{act}}(\boldsymbol{W}, \widetilde{\boldsymbol{W}}) = \mathrm{Tr}\Big((\boldsymbol{W} - \widetilde{\boldsymbol{W}})^\top \boldsymbol{\Sigma}_x (\boldsymbol{W} - \widetilde{\boldsymbol{W}})\Big) \tag{14}$$

Thus activation-aware compression effectively replaces the Euclidean metric $\langle \boldsymbol{A}, \boldsymbol{B}\rangle_F = \mathrm{Tr}(\boldsymbol{A}^\top \boldsymbol{B})$ by a weighted inner product $\langle \boldsymbol{A}, \boldsymbol{B}\rangle_{\boldsymbol{\Sigma}_x} = \mathrm{Tr}(\boldsymbol{A}^\top \boldsymbol{\Sigma}_x \boldsymbol{B})$ that emphasizes directions with large activation variance. Methods such as ASVD and SVD-LLM can be interpreted as performing low-rank approximation in this weighted geometry, often via whitening transformations that absorb $\boldsymbol{\Sigma}_x^{1/2}$ into the weight matrices before applying SVD.

**What an activation-aware Tucker objective would require.** In LeSTD, the MHA parameters of one layer are reshaped into a 4th-order tensor $\boldsymbol{\mathcal{W}}_{\mathrm{total}} \in \mathbb{R}^{d_{\mathrm{model}} \times d_{\mathrm{head}} \times 4 \times h}$, and Stage I solves the Tucker problem

$$\min_{\boldsymbol{\mathcal{G}}_{\mathrm{total}}, \boldsymbol{U}^{(1)}, \boldsymbol{U}^{(2)}, \boldsymbol{U}^{(3)}} \big\|\boldsymbol{\mathcal{W}}_{\mathrm{total}} - \boldsymbol{\mathcal{G}}_{\mathrm{total}} \times_1 \boldsymbol{U}^{(1)} \times_2 \boldsymbol{U}^{(2)} \times_3 \boldsymbol{U}^{(3)}\big\|_F^2 \tag{15}$$

subject to $(\boldsymbol{U}^{(n)})^\top \boldsymbol{U}^{(n)} = \boldsymbol{I}$ and shared $\boldsymbol{U}^{(n)}$ across all heads, as detailed in Section 3.1. To make this objective activation-aware, one would have to replace the Frobenius norm in Eq. (15) by a tensor analogue of the weighted metric Eq. (14). However, unlike the single-matrix case, an MHA block couples: (i.) multiple types of projections ($Q/K/V/O$), (ii.) multiple heads, and (iii.) multi-token activations flowing through attention. Formally, if we denote by $\boldsymbol{\mathcal{X}}$ the joint tensor of pre-activation features feeding into all MHA projections, the natural activation-aware loss would involve an expectation of the form

$$\mathcal{L}_{\mathrm{act}}^{\mathrm{Tucker}} = \mathbb{E}_{\boldsymbol{\mathcal{X}}}\left[\big\|f_{\boldsymbol{\mathcal{W}}_{\mathrm{total}}}(\boldsymbol{\mathcal{X}}) - f_{\widetilde{\boldsymbol{\mathcal{W}}}_{\mathrm{total}}}(\boldsymbol{\mathcal{X}})\big\|_F^2\right] \tag{16}$$

where $f_{\boldsymbol{\mathcal{W}}}$ denotes the linear part of the MHA block parametrized by $\boldsymbol{\mathcal{W}}$. Even under strong linearization assumptions (ignoring softmax and non-linearities), this expectation induces a 4th-order covariance operator over the tensorized weights:

$$\mathcal{L}_{\mathrm{act}}^{\mathrm{Tucker}} = \langle \boldsymbol{\mathcal{W}}_{\mathrm{total}} - \widetilde{\boldsymbol{\mathcal{W}}}_{\mathrm{total}}, \mathcal{C}(\boldsymbol{\mathcal{W}}_{\mathrm{total}} - \widetilde{\boldsymbol{\mathcal{W}}}_{\mathrm{total}})\rangle \tag{17}$$

where $\mathcal{C}$ is a positive-definite linear operator that depends on the joint statistics of $Q/K/V/O$ activations, heads, and tokens. To retain the shared-factor structure of Eq. (3), one would need $\mathcal{C}$ to factorize approximately as a Kronecker product over modes (e.g., $\boldsymbol{\Sigma}^{(1)} \otimes \boldsymbol{\Sigma}^{(2)} \otimes \boldsymbol{\Sigma}^{(3)}$), and these mode-wise covariances would have to be compatible with a single set of orthogonal factors $\boldsymbol{U}^{(1)}, \boldsymbol{U}^{(2)}, \boldsymbol{U}^{(3)}$ shared across all heads and projections. In practice, $Q/K/V/O$ activations have quite different distributions and scaling, and the covariance is neither separable nor shared across heads. Therefore, there is no obvious way to define a single activation-weighted metric under which all rank-1 basis tensors remain mutually orthogonal.

**Impact on Stage II and sparse-core theory.** Stage II of LeSTD crucially relies on the fact that, under the Frobenius inner product, the rank-1 basis tensors $\{\boldsymbol{B}_\beta\}$ induced by $\boldsymbol{U}^{(1)}, \boldsymbol{U}^{(2)}, \boldsymbol{U}^{(3)}$ form an orthonormal system: $\langle \boldsymbol{B}_\beta, \boldsymbol{B}_\gamma \rangle_F = \delta_{\beta\gamma}$. This orthonormality allows us to express the reconstruction error as

$$\|\boldsymbol{\mathcal{W}}_{\mathrm{total}} - \widehat{\boldsymbol{\mathcal{W}}}_{\mathrm{total}}\|_F^2 = \|\boldsymbol{R}\|_F^2 + \sum_\beta g_\beta^2$$

and to show that zeroing a single coefficient $g_\beta$ increases the error by exactly $g_\beta^2$ (Eq. (5)). Hence the optimal $k$-term approximation is obtained by hard-thresholding on $|g_\beta|$, and the refitting step admits a one-dimensional closed form (Eq. (6)). If switch to a general activation-weighted metric induced by $\mathcal{C}$, the basis $\{\boldsymbol{B}_\beta\}$ is no longer orthonormal:

$$\langle \boldsymbol{B}_\beta, \boldsymbol{B}_\gamma \rangle_{\mathcal{C}} = \langle \boldsymbol{B}_\beta, \mathcal{C}(\boldsymbol{B}_\gamma) \rangle_F \neq \delta_{\beta\gamma}$$

The error becomes

$$\|\boldsymbol{\mathcal{W}}_{\text{total}} - \widehat{\boldsymbol{\mathcal{W}}}_{\text{total}}\|_{\mathcal{C}}^2 = \sum_{\beta,\gamma} g_\beta g_\gamma \langle \boldsymbol{B}_\beta, \mathcal{C}(\boldsymbol{B}_\gamma) \rangle_F$$

and the increase in loss associated with pruning a single $g_\beta$ depends on all cross-terms with $\gamma \neq \beta$. In this regime: (i.) magnitude-based ordering of $|g_\beta|$ is no longer the optimal, (ii.) identifying the best $k$ coefficients is a coupled combinatorial problem, and (iii.) refitting remaining coefficients no longer reduces to independent one-dimensional least squares. One can, in principle, define an activation-weighted sparse Tucker objective and derive generalized normal equations, but doing so would forfeit the key advantages of Stage II: data-free operation, closed-form importance scores, and efficient pruning with guaranteed monotone error accounting.

**Practical considerations and our choice in this work.** From a practical standpoint, activation-aware Tucker compression would also require: (i.) storing and streaming layer-wise activation tensors for each MHA block during calibration, (ii.) estimating high-order covariances (or at least their low-rank surrogates), and (iii.) calibrating separately for each model, dataset, and even downstream task. Recent empirical studies on PTQ and pruning show that compression quality can be highly sensitive to the choice of calibration corpus (Frantar et al., 2022; Lin et al., 2024; Shang et al., 2023), and that calibration on mismatched data can even hurt generalization to new tasks. In contrast, LeSTD is fully data-free and immediately applicable to any pretrained checkpoint without additional data curation or repeated activation collection.

In summary, while in principle one could attempt, but under our design constraints this is practically infeasible to define an activation-weighted inner product and to formulate an activation-aware sparse Tucker problem, such a design would:

1. break the orthonormal structure that underpins our closed-form importance scores and refitting updates;
2. require strong and arguably unrealistic separability assumptions on multi-head, multi-projection activation statistics to preserve shared factors;
3. sacrifice the data-free nature of LeSTD by introducing the non-trivial calibration overhead and dataset dependence.

## K  LLM USAGE

During the preparation of this paper, we used OpenAI's ChatGPT-5 to assist with language polishing and improving readability of the manuscript. In addition, we used OpenAI's Codex to help parse and understand baseline implementations by reading and decomposing publicly available source code. No part of the core technical contributions, experimental design, or results analysis was generated by LLMs; all scientific ideas, methodology, and validation remain the sole work of the authors.

