# OpenReview forum: "LeSTD: LLM Compression via Learning-based Sparse Tensor Decomposition"
_ICLR.cc/2026/Conference — ICLR 2026 Poster_

### Official Review · Reviewer_W25T · 2025-11-01

**Soundness:** 3
**Presentation:** 3
**Contribution:** 3
**Rating:** 6
**Confidence:** 4

**Summary:**

The paper introduces LeSTD (Learning-based Sparse Tensor Decomposition), a data-free, post-training compression framework for large language models. It addresses the dense core bottleneck in tensor decomposition methods by learning a shared basis across attention heads and then applying a theoretically grounded pruning mechanism to create an ultra-sparse core tensor. This yields higher compression ratios without accuracy loss. LeSTD performs inference directly in the compressed domain, providing throughput gains without custom kernels.

**Strengths:**

1. If I understand correctly, prior methods such as SVD-LLM and ASVD require calibration data, whereas LeSTD operates entirely data-free—yet still outperforms them. This is quite impressive.
2. The idea of tensorizing all weights into a unified 4D structure and applying Tucker decomposition** is both intuitive and novel, offering a principled way to exploit inter-layer correlations that matrix-based methods overlook.

**Weaknesses:**

Check in questions part.

**Questions:**

1. The authors should clarify the distinction between LeSTD and TensorLLM. If my understanding is correct, Section 3.1 is identical with TensorLLM, while Section 3.2 is the main difference. It would be helpful to make this relationship explicit.

2. As I understand, LeSTD does not require calibration data, which is a key advantage. Would it be possible to incorporate activation information into this framework to further improve performance? I suspect this might be challenging under Tucker decomposition, but a discussion on its feasibility would be valuable.

3. The paper could include comparisons with more advanced SVD-based pruning methods, such as Basis Sharing [1] and Pivoting Factorization [2]. The concept of Basis Sharing appears somewhat related, since Tucker decomposition also captures inter-weight similarity.

4. It would strengthen the presentation to include an inference algorithm, similar to Algorithm 1 for pruning. Section 3.3 is currently somewhat difficult to follow, and a concise pseudocode description would improve clarity.

5. Will the code be released? If I understand correctly, the original linear layer requires one matrix multiplication, low-rank layers require two, and the proposed Tucker-based structure requires four. This may introduce additional I/O overhead. The authors claim that the method achieves speedup without custom kernels—i.e., purely using PyTorch—which is an impressive claim but also raises concerns. Providing the implementation during the rebuttal period would greatly improve credibility and reproducibility.

[1] Wang, Jingcun, et al. "Basis Sharing: Cross-Layer Parameter Sharing for Large Language Model Compression." The Thirteenth International Conference on Learning Representations.
[2] Zhao, Jialin, Yingtao Zhang, and Carlo Vittorio Cannistraci. "Pivoting Factorization: A Compact Meta Low-Rank Representation of Sparsity for Efficient Inference in Large Language Models." Forty-second International Conference on Machine Learning.

---

> ### Author Response · Authors · 2025-11-23
> **Part One**
>
> We thank the reviewer for this insightful question, which allows us to better clarify our design. We address the reviewer's questions below.
> ### Question 1: Distinction between LeSTD and TensorLLM
> Although LeSTD and TensorLLM use a similar MHA tensor layout, their compression and execution mechanisms are fundamentally different. LeSTD applies Tucker factorization only to the first three modes while keeping the head mode (mode 4) fixed (as shown in Eq. (3)), preserving head-specific information. This design enables LeSTD to achieve higher accuracy than TensorLLM at comparable compression ratios. Furthermore, LeSTD’s Stage II introduces a sparsification and refitting procedure that produces an extremely sparse core tensor, an aspect not present in TensorLLM. Moreover, associated with StageI&II, MHA computations are performed directly in the compressed domain (i.e., $\mathcal{G}_{\text{sparse}}$ and $\mathbf{U}^{(1/2/3)}$). As a result, no decompression step is required, which further improves inference efficiency.
>
> More detailed distinctions on goal and mechanism are shown below:
> - Stage I as a building block, not final method. In LeSTD, the shared-subspace Tucker decomposition in Section 3.1 is only Stage I: it provides a convenient common low-rank subspace in which all heads and projections live. We then treat the Tucker core as a new compression target, rather than as the final representation (this is what TensorLLM does).
> - Stage II as importance-based sparse core learning (Section 3.2). TensorLLM leaves the Tucker cores dense and only controls compression by choosing the multilinear ranks. However, LeSTD introduces a second stage that (i) derives a closed-form importance score for each core coefficient, (ii) iteratively prunes the least important entries with a pruning rate $\alpha$, and (iii) refits the remaining coefficients in closed form (Eq.(5) – Eq.(6), and Algorithm 1). This yields an ultra-sparse core tensor with explicit Frobenius-error control, which does not appear in TensorLLM.
> - Inference directly in compression domain (Section 3.3). TensorLLM reconstructs dense attention weights from the Tucker factors and then runs the standard dense MHA computations. While LeSTD instead never materialises the dense $\mathbf{W}^{Q}, \mathbf{W}^{K},\mathbf{W}^{V},\mathbf{W}^{O}$: we (i) project activations once into the shared subspace, (ii) contract the sparse core into small per-head matrices $\mathbf{M}[i,t]$, and (iii) compute all $Q/K/V/O$ projections via sparse–dense products (Algorithm 2). This directly reduces arithmetic complexity and improves measured *tokens/sec*, which is outside the scope of TensorLLM.

---

> ### Author Response · Authors · 2025-11-23
> **Part Two**
>
> ### Quesiton 2: The feasibility of incorporating activation information
> It is possible to incorporate activation information into the LeSTD framework, but doing so is non-trivial. Our current design relies on a shared Tucker decomposition with an orthonormal latent basis and a data-free sparse core, which makes direct integration of activation statistics less straightforward. We have added a dedicated discussion of this feasibility question and potential extensions in **Appendix J** of the revised manuscript. A detailed explanation is provided below.
>
> In the matrix setting, activation-aware methods such as ASVD and SVD-LLM effectively replace the standard Frobenius metric by a covariance-weighted inner product and then perform low-rank approximation in that weighted geometry. In LeSTD, however, we first reshape an entire MHA block into a 4th-order tensor and learn shared orthogonal factors across heads and across $Q/K/V/O$, and then perform magnitude-based sparsification and refitting in the resulting orthonormal latent space. Our closed-form importance scores and refitting step (Stage II) rely critically on this orthonormal structure: pruning a single core coefficient has an exactly local effect on the Frobenius reconstruction error, which makes the optimal $k-$term approximation equivalent to hard-thresholding by $\vert g_{\beta} \vert$ and and yields one-dimensional refitting updates.
>
> If we were to make the Tucker objective activation-aware, we would need to define a tensor analogue of a covariance-weighted metric that couples (i) multiple projections ($Q/K/V/O$), (ii) multiple heads, and (iii) multi-token activations. Under such a weighted metric, the latent basis is no longer orthonormal: the loss becomes a fully coupled quadratic form over all core coefficients, magnitude ordering is no longer optimal, and refitting no longer decomposes into independent one-dimensional problems. In other words, we would lose exactly the properties that make Stage II efficient, data-free, and theoretically clean. In addition, an activation-aware Tucker variant would require storing and streaming high-order activation tensors, estimating layer-wise covariances (or surrogates), and calibrating separately for each model/dataset/task, which sacrifices the main practical advantage of LeSTD: being immediately applicable to any checkpoint without calibration data.
>
> For these reasons, we deliberately keep LeSTD fully data-free in this work. We agree that designing approximate activation-aware extensions, for example, by imposing structured assumptions on activation statistics or using activations only to tune ranks / pruning schedules rather than the full Tucker objective, is an interesting direction for future work.
>
> ### Question 3: Compared with Basis Sharing and Pivoting Factorization
> According to the comparison with basis sharing and pivoting factorization, we have added a dedicated comparison and discussion in **Appendix E.5 (Related Works)** and **Appendix H (Experimental Results)**, together with new experiments on GPT-J on WikiText-2.
>
> | Method        | r = 0.2 Perplexity($\downarrow$) | r = 0.2 Tokens/sec($\uparrow$) | r = 0.8 Perplexity($\downarrow$) | r = 0.8 Tokens/sec($\uparrow$) |
> |--------------|------------------------|-------------------------|------------------------|-------------------------|
> | Basis Sharing | 99.16                  | 14025.69               | 12.30                  | 10091.19                |
> | MPIFA         | 102.15                 | 14086.19               | 12.88                  | 9488.84                 |
> | LeSTD (Ours)  | 80.37                  | 15636.24               | 8.92                   | 11514.42                |
>
> Empirically, the table above (Table 7 in the revised paper) reports a end-to-end comparison at two global compression ratios $r\in \{0.2,0.8\}$. At the aggressive setting $r=0.2$, all methods suffer, but Basis Sharing and MPIFA reach perplexities of 99.16 and 102.15 with $\sim14K$ tokens/sec throughput, whereas LeSTD achieves a lower perplexity of 80.37 and higher throughput of 15,636 tokens/sec. At the milder setting $r=0.8$, Basis Sharing and MPIFA incur noticeable accuracy drops (perplexity 12.30 and 12.88), while LeSTD attains perplexity 8.92 with improved throughput (11,514 tokens/sec). Overall, these results indicate that LeSTD is competitive with, and often stronger than, these advanced SVD-based pruning baselines, offering a better accuracy–efficiency trade-off under comparable global compression ratios.

---

> ### Author Response · Authors · 2025-11-23
> **Part Three**
>
> ### Question 4: A pseudocode to improve readability
> We have added a pseudocode for the inference algorithm: **Algorithm 2: Inference Without Reconstruction in LeSTD**, at the end of **Section 3.3**.
>
> The pseudocode summarizes the key steps of compressed-domain inference, including the shared left projection, core contraction into per-head matrices, $Q/K/V$ projection, attention, output projection.
>
> ### Question 5: Code Release
> In the revised manuscript we have added a **Reproducibility Statement** that includes an anonymous code repository link.

---

### Official Review · Reviewer_ee46 · 2025-11-01

**Soundness:** 3
**Presentation:** 2
**Contribution:** 2
**Rating:** 4
**Confidence:** 3

**Summary:**

This paper proposes a two-stage large language model compression framework based on Tucker decomposition. The first stage performs subspace decomposition to obtain low-rank latent representations, while the second stage compresses the core tensor via a closed-form sparse pruning method. The paper is well-organized, with sound theoretical analysis, detailed mathematical derivations, and extensive experimental validation.

**Strengths:**

1.	The paper introduces a concise and elegant closed-form sparsification method in the decomposed latent space.
2.	The mathematical derivations are thorough and rigorous, providing solid theoretical support for amplitude pruning in the Tucker latent space.
3.	The motivation is clearly articulated, and the background is well presented.

**Weaknesses:**

1.	There are some formatting issues in the manuscript (e.g., line 100 references Figure 4 located at line 227).
2.	The figures and their explanations are somewhat unclear. For example, in Figure 3, the components are scattered without clear annotation or explanation of what each parameter represents.
3.	The introduction of the core tensor compression method—the paper’s key innovation—is somewhat disorganized. Although the mathematical derivations are complete, their presentation could be improved with more structured figures and explanations.

**Questions:**

In the throughput analysis, the paper claims inference acceleration using standard PyTorch functions on sparse cores. However, it is uncertain whether PyTorch natively accelerates unstructured sparsity. Have the authors considered other possible factors that might contribute to the observed speedup?

---

> ### Author Response · Authors · 2025-11-23
> **Part One**
>
> We thank the reviewer for pointing out these potential improvements, they have been very helpful in strengthening the paper. We address the reviewer's weaknesses and questions below.
>
> ### Weakness 1: Formatting issue
> We thank the reviewer for pointing this out. We have revised the manuscript accordingly. For example, in Section 1, the description of the Dense Core Bottleneck has been updated: we removed the forward reference to a later figure and rewrote the passage as a clear, standalone explanation to improve readability.
>
> ### Weakness 2: Clear Annotation
> We have revised the figures and their accompanying explanations to improve readability. In particular, Figure 3 has been updated with a new legend to clarify the semantics of each component. The caption has also been expanded to provide more detailed explanations of the involved parameters.
>
> ### Weakness 3: Improve the introduction of the core tensor compression method
> We appreciate the reviewer’s comment and have improved the presentation of the core-tensor compression method accordingly. Specifically:
> 1. We revised the text between Section 3 and Section 3.1 to include additional high-level explanation, a roadmap of the subsequent subsections, and a clearer description of their logical relationships.
> 2. In Section 3.3 (“Inference Without Reconstruction”), we added pseudocode to more transparently illustrate the end-to-end workflow.
>
> ### Question 1: Throughput Analysis
> For sparse matrix multiplication, we used `torch.matmul` in our main experiments to ensure a fair comparison and to avoid any unintended speedup from specialized sparse kernels. Importantly, torch.matmul does not accelerate unstructured sparse matrices, so it provides a neutral baseline for evaluating our method.
>
> To examine the potential impact of more optimized kernels, we also implemented a variant using `torch.sparse.mm`. As shown in the table below, on GPT-J with WikiText-2 at an MHA compression ratio of $\rho = 0.6$, the throughput increases from 11,945.98 to 12,592.54 tokens/sec ($\approx$ 5% improvmenet). This indicates that while the current speedup primarily comes from LeSTD’s structural compression, additional gains are possible with more optimized sparse kernels, suggesting room for future engineering improvements.
>
>
>
>    | Kernel on sparse cores          | Throughput (tokens/sec) | Relative speed |
>    |---------------------------------|-------------------------|----------------|
>    | `torch.matmul`                  | 11945.98                | --             |
>    | `torch.sparse.mm`              | 12592.54                | 1.05×          |
>
>
> Secondly, regarding other potential factors contributing to the observed speedup: several effects may arise from operating on significantly smaller compressed representations, such as reduced data movement between GPU memory and GPU units, fewer matrix-multiplication API calls, lower decompression overhead, and improved cache locality due to smaller data. These effects are inherent consequences of the high compression ratio achieved by our method.

---

### Official Review · Reviewer_y5qd · 2025-11-01

**Soundness:** 3
**Presentation:** 3
**Contribution:** 2
**Rating:** 6
**Confidence:** 4

**Summary:**

In this submission a novel post-training compression framework for LLM called LeSTD is proposed. The approach is two-step: in the first step Tucker decomposition is applied to learn shared orthonormal factors across attention heads for each layer of the LLM. During the second step the ultra-sparse core tensor is created using importance-based iterative pruning. The importance score is derived from a reconstruction error.

**Strengths:**

1.  The two-step approach is well-motivated and intuitive: first we learn a basis and then we sparsify within that basis
2.  Theoretical justification for magnitude-based pruning is solid.
3.  Extensive experiments with different models and different datasets.
4.  LeSTD shows improvement over the competing methods across different compression ratios.
5.  Paper is well-written, the presentation of the method is good with neat illustration;
6.  The limitations are acknowledged and discussed (unstructured sparsity and MHA-only compression)

**Weaknesses:**

0.  The overall novelty of the submission is limited: Stage I is a known combination of the Tucker decomposition and HOOI, Stage II reduces to magnitude pruning (which is known optimal for orthonormal bases). The contribution is primarily in combining these for LLMs compression rather than theoretical/methodological novelty

1.  Other sparse tensor methods are not considered (CP decomposition, Tensor-Train Decomposition, structured/block-sparse Tucker variants, etc)
2.  Experiments do not include statistical significance (which is crucial in such works): no error bars, confidence intervals, or multiple runs reported;

3.  Limited ablation studies: only 6 rank configurations tested (Table 2), no ablation on pruning rate α, refitting frequency, or HOOI convergence criteria

4.  For the throughput, LeSTD sometimes loses to SVD-LLM (e.g., for OPT-30B on MathQA at 0.8 compression rate), but analysis of when/why is insufficient

**Questions:**

1.  Can you provide error bars across multiple experimental runs to assess statistical significance of the improvements over other methods?
2.  How sensitive is performance to the pruning rate α? Only α=0.1 is mentioned; was this tuned?
3.  What is the actual sparse indexing overhead at different sparsity levels? How does this affect real compression ratios?
4.  Why does LeSTD sometimes lose to SVD-LLM on throughput (e.g., OPT-30B)? Can you characterize when your method wins vs. loses?
5.  Can you provide results extending LeSTD to FFN layers, even if preliminary?
6.  How important is the refitting step (Eq. 6)? How does skipping this step affect the performance?
7.  What are the wall-clock compression times for Stage I and Stage II compared to baselines?

---

> ### Author Response · Authors · 2025-11-23
> **Part One**
>
> We sincerely thank the reviewer for the thoughtful comments and constructive feedback on our work. We address the reviewer's weaknesses and questions below.
>
> ### Weakness 0: Novelty clarification
> We appreciate the reviewer’s observation regarding the classical nature of the building blocks used in our method. We would like to clarify that our goal is not to claim novelty at the level of these primitives. Rather, the contribution lies in how these components are specialized, integrated, and scaled into a practical and theoretically grounded compression framework for modern LLMs. In particular: In particular:
> - From Tucker for MHA to a two-stage, high-ratio framework. TensorLLM (which is also one of our baselines) has recently shown that stacking all MHA weights into a single tensor can yield a more compact representation, and thus higher compression ratios, than matrix-by-matrix compression. Nevertheless, this representation incurs an inherent bottleneck, for example the dense-core problem we identify in this paper: when using a standard Tucker decomposition, the shared core tensor remains dense, which fundamentally limits both the achievable compression rate and the actual computational savings. Stage II is explicitly designed to remove the bottleneck: we derive an importance score in the Tucker core, prove that best$-k$  approximation can be obtained by keeping the largest $\vert g_{\beta} \vert$, and use closed-form refitting to iteratively push the core into the extreme sparse regime.
> - A tailor-made pipeline for MHA (and further FFN (please also refer to the response to Question 5 and Reviewer RLPz's Weakness 2)). Another novelty of LeSTD is that, we build a two-stage pipeline for Transformer blocks, and propose a direct inference scheme that operates entirely in the compressed domain without reconstructing the original weights.
>
> ### Weakness 1: Differnet Tensor Decomposition Format
> According to the other sparse tensor methods, we have added the discussion about CP, Tensor Train (TT), and structured / block-sparse Tucker variants, and why we adopt a Tucker-with-sparse-core topology for LeSTD in **Appendix E.4**.
>
> - **CP v.s. Tucker.** Appendix E.4 summarizes classical results showing that CP often needs substantially larger rank than Tucker to reach a comparable approximation error and that CP fitting is prone to degeneracy/ill-conditioning on complex or nearly collinear data. In our setting, we construct a 4-way joint MHA tensor whose modes encode (model dimension, head dimension, projection type $Q/K/V/O$, head index), our design goal is to learn shared subspaces along the first three modes while keeping per-head variation in a small core. We now explicitly show that a Tucker model:
> $$
> \mathbf{\mathcal{W}} \approx \mathbf{\mathcal{G}}\times_1 \mathbf{U}^{(1)} \times_2 \mathbf{U}^{(2)} \times_3 \mathbf{U}^{(3)} \times_4 \mathbf{U}^{(4)}
> $$
> with $\mathbf{U}^{(1)},\mathbf{U}^{(2)},\mathbf{U}^{(3)}$ shared and mode-4 slices of $\mathbf{\mathcal{G}}$ acting as per-head parameters, matches this pattern directly. In contrast, a CP model would express $\mathbf{\mathcal{W}}$ as a sum of rank-1 components that simultaneously couple all four modes, maintaining head-specific structure either requires tying the head factors (hurting expressiveness) or significantly increasing the CP rank (hurting compression).
> - **TT v.s. Tucker.** Appendix E.4 further explains that TT is particularly appealing for very high-order tensors with many small modes, where the long chain topology is natural. Our joint MHA tensor is instead low-order (4 modes) but with semantically structured dimensions: model dimension, head dimension, projection type, and head index. Mapping it into a TT chain would require non-trivial reshaping and ordering choices and makes it less clear how to (i) enforce a simple shared subspace across heads, and (ii) keep per-head cores with a straightforward interpretation.
> - **Structured/Block-sparse Tucker Variants.** As we response to Reviewer RLPz's Question 2, our current design uses **element-wise** important score on the core: for each coefficient $\vert g_{\beta} \vert$, we can compute exactly how much the reconstruction error would increase if we set $g_{\beta}=0$, and we greedily prune the least important entries. Because this operates at the finest granularity, it tends to preserve accuracy better for a given sparsity level. Compared to structural/block-sparse Tucker, we propose to use element-wise scheme, this is a design trade-off: structured patterns are more friendly to specific hardware and specialized kernels, while usually incurs some additional accuracy loss at the same overall sparsity level compared to our current element-wise scheme.

---

> ### Author Response · Authors · 2025-11-23
> **Part Two**
>
> ### Weakness 2: Statistical Significance
> Regarding the statistical significance, we have added statistical significance analyses in a new appendix (**Appendix C.4**). For the detailed results and discussion, please refer to our response to Question 1.
>
> ### Weakness 3: More Comprehensive Ablation Study
> For more comprehensive ablation study, we have added/revised multiple appendices, which provide:
> - More rank configuration (revised **Appendix C.1**). In the revised manuscript, we have expanded the rank-sensitivity study in **Appendix C.1** by adding more Tucker-rank configurations for GPT-J on WikiText-2 at an overall MHA compression ratio around 0.6. The updated Table 2 now includes 8 representative (R1, R2, R3) settings, covering a reasonably wide range of ranks and compression ratios:
> | R1   | R2  | R3 | Ratio | Perplexity($ \downarrow$) |
> |------|-----|----|-------|--------------|
> | 4096 | 256 | 4  | 1.00  |  8.86        |
> | 1536 | 320 | 4  | 0.56  | 13.72        |
> | 1536 | 384 | 4  | 0.66  | 11.91        |
> | 1792 | 320 | 4  | 0.65  | 11.33        |
> | 2048 | 224 | 4  | 0.56  | 10.53        |
> | 2048 | 256 | 4  | 0.63  |  9.53        |
> | 1920 | 256 | 4  | 0.60  |  9.92        |
> | 2048 | 288 | 4  | 0.66  |  9.26        |
>
> As discussed in **Appendix C.1**, the trends are consistent and clear:
> (a.) Increasing $R1$ or $R2$ monotonically improves perplexity at a given compression level;
> (b.) Configurations such as $(R1=2048, R2=256, R3=4)$ and $(R1=2048, R2=288, R3=4)$ reach near-original perplexity (down to $9.26$) at compression ratios around $0.63 – 0.66$;
> (c.) Smaller ranks (e.g., $R1=1536$, $R2=320$) lead to noticeably worse perplexity.
>
> Overall, this expanded confirms that our reported rank choice in the main experiments lies in a stable regime that balances model size and accuracy, and that the conclusions are not tied to a single or a very small set of rank configurations.
>
> - Pruning rate $\alpha$ (Added **Appendix C.5**): For detailed repsonse, please refer to Question 2.
> - Importance of Refitting (Added **Appendix C.6**): For detailed response, please refer to Question 6.
> - HOOI Convergence Criteria: In our implementation, we run higher-order orthogonal iteration (HOOI) with a standard convergence criterion: we terminate when either (i) Reconstruction error (Eq.(4)) falls below $10^{-2}$, or (ii) the number of iterations reaches 50, whichever comes first. This tolerance-plus-max-iterations rule is consistent with common HOOI practice in the tensor-decomposition literature, where iterations are stopped once the reconstruction error changes only marginally or a fixed iteration budget is exhausted[1,2,3].
>
> ### Weakness 4: Some insufficient compared to SVD-LLM
> Please refer to the response to Question 4.
>
> ### Question 1: Error bar with mutliple runs
> We have conducted repeated runs of key experiments to measure variance: On GPT-J/WikiText-2, we ran LeSTD, SVD-LLM, and TensorLLM 10 times with different random seeds and measure task performance. New results have been added in **Appendix C.4**.
>
> | Method        | Perplexity($\downarrow$)       | Tokens/sec($\uparrow$)        |
> |--------------|----------------------|------------------------|
> | SVD-LLM      | 20.28 ± 0.14         | 11105.18 ± 90.65       |
> | TensorLLM    | 9.87 ± 0.10          | 8572.11 ± 103.18       |
> | LeSTD (Ours) | 9.48 ± 0.09          | 12025.90 ± 91.77       |
>
> Concretely, at a fixed compression ratio of 0.6 we report mean $\pm$ standard deviation for both perplexity and throughput: SVD-LLM obtains $20.28 \pm 0.14$ perplexity and $11105.18 \pm 90.65$ tokens/sec, TensorLLM achieves $9.87 \pm 0.10$ and $8572.11 \pm 103.18$, while LeSTD reaches $9.48 \pm 0.09$ and $12025.90 \pm 91.77$. The variance across runs is small for all methods, and the gaps between LeSTD and TensorLLM/SVD-LLM in both perplexity and throughput are substantially larger than one standard deviation. This indicates that our improvements are stable across seeds and unlikely to be artifacts of a particular random run.
>
> ### Reference
> [1] De Lathauwer, Lieven, Bart De Moor, and Joos Vandewalle. "On the best rank-1 and rank-(r 1, r 2,..., rn) approximation of higher-order tensors." SIAM journal on Matrix Analysis and Applications 21.4 (2000): 1324-1342.
>
> [2] Kolda, Tamara G., and Brett W. Bader. "Tensor decompositions and applications." SIAM review 51.3 (2009): 455-500.
>
> [3] Liu, Siqi, Xiaoyu Shi, and Qifeng Liao. "Rank-adaptive tensor completion based on Tucker decomposition." Entropy 25.2 (2023): 225.

---

> ### Author Response · Authors · 2025-11-23
> **Part Three**
>
> ### Question 2: Pruning rate $\alpha$ sensitivity
> According to the sensitivity of pruning rate $\alpha$, we would like to clarify that, in LeSTD, $\alpha$ only controls how aggressively the scheme prunes per iteration in Stage II, but it does not fix the final sparsity of the shared core tensor $\mathbf{\mathcal{G}}$, which is determined by the target sparsity $S_{\text{target}}$ derived from the desired compression ratio. Given $k$ non-zero coefficients, each iteration removes $\lceil \alpha \cdot k\rceil$ entries, followed by a closed-form refitting step; thus α trades off the number of iterations versus the aggressiveness of each step, rather than directly setting the final sparsity pattern.
>
> | Pruning rate α | Perplexity($\downarrow$) (WikiText-2) |
> |----------------|----------------------------|
> | 0.05           | 9.49                       |
> | 0.10           | 9.50                       |
> | 0.15           | 9.51                       |
> | 0.20           | 9.52                       |
> | 0.25           | 9.53                       |
> | 0.30           | 9.54                       |
> | 0.35           | 9.60                       |
> | 0.40           | 9.67                       |
> | 0.45           | 9.75                       |
> | 0.50           | 9.82                       |
>
>
> To assess sensitivity, we fix the target core sparsity for GPT-J on WikiText-2 at an overall compression ratio of 0.6 and sweep $\alpha$ from 0.05 to 0.50 in steps of 0.05. As reported in **Appendix C.5 (Figure. 8)**, LeSTD is very robust to the choice of α in a broad range: for $\alpha \in [0.05, 0.30]$, the WikiText-2 perplexity stays within a narrow band of 9.49–9.54, indicating that our importance-based pruning and refitting procedure consistently converges to nearly the same solution under different per-iteration pruning rates. When α is made more aggressive (e.g., $\alpha \ge 0.35$), the perplexity gradually increases (reaching 9.82 at $\alpha = 0.50$), which we attribute to pruning a larger fraction of coefficients before the refitting step can fully readjust the remaining ones. Importantly, even in this regime, LeSTD remains noticeably better than matrix-wise SVD and dense-core Tucker at the same compression ratio.
>
> In our main experiments, we therefore use a single default value $\alpha = 0.1$, chosen from this flat region without further tuning per model/dataset.

---

> ### Author Response · Authors · 2025-11-23
> **Part Four**
>
> ### Question 3: Sparse Indexing Overhead
> We thank the reviewer for asking this practical quesiton. We break down the answer into following points:
> - How we count compression, and consistency with baselines?
>
> As clarified in **Appendix G**, our reported global compression ratio $\rho$ is defined purely in terms of parameter values (i.e., number of stored scalars), following the same convention as the baselines we compared. Concretely, for each MHA block we count: $P _ {\text{compressed}} = d _ {\text{model}}\cdot R _ 1 + d _ {\text{head}}\cdot R _ 2 + 4R _ 3 + nnz(\mathcal{G} _ {\text{sparse}})$ and set $\rho=P_{\text{compressed}}/P_{\text{original}}$, where $P_{\text{original}} = 4d^2_{\text{model}}$ is the parameter count of the four dense projection matrices in the original MHA. As noted in the paper, this deliberately ignores sparse index overhead, so that the accounting is directly comparable to baselines that also do not model storage-format metadata (i.e., indexing overhead).
>
> - Actual sparse indexing overhead in our implementation.
>
> In our implementation, the sparse core $\mathcal{G} _ {\text{sparse}}$ is stored in COO format via `torch.sparse_coo_tensor`. COO stores the non-zeros as (indices, values) pairs: an indices tensor of shape $(d,nnz)$  with integer coordinates and a values tensor of shape $(nnz)$ with floating-point entries, where $d$ is the tensor order. For a given core of total size $N$ elements and density $p=nnz/N$, the storage cost of COO can be written as $\text{Cost} _ {\text{COO}}\approx nnz\cdot (b _ {\text{val}}+d\cdot b _ {idx})= Npb _ {\text{idx}}(1+db _ {\text{idx}}/b _ {\text{val}})$, where $b_{\text{val}}$ and $b_{\text{idx}}$ are the bytes used for a value and an index respectively. A dense representation of the same core would cost $\text{Cost} _ {\text{dense}}=Nb _ {\text{val}}$. Thus the relative storage overhead of COO vs. dense for the same core shape is $\frac{\text{Cost} _ {\text{COO}}}{\text{Cost} _ {dense}}=p(1+db _ {\text{idx}}/b _ {\text{val}})$. With FP16 values and 64-bit integer indices (PyTorch’s default), we have $b_{\text{val}}=2$ bytes and $b_{\text{idx}}=8$ bytes, so $b_{\text{idx}}/b_{\text{val}}=4$, this give a factor of $1+3\times 4=13$ for the 3D core tensor. This implies that, the break-even density is roughly $p\approx 1/13 \approx 0.077$ (i.e., 0.923 sparsity). In the regimes we operate in, Stage II prunes the shared core to extremely high sparsity (single-digit percent densities) in order to hit the target $ \rho$. Under such densities, the COO representation of $\mathcal{G}_{\text{sparse}}$ remains substantially smaller than a dense core of the same shape, even after accounting for indices.
>
> - Impact on real compression rate
>
> To evaluate the “true” compression ratio including index overhead , this can be done by augmenting $P_{\text{compressed}}$ with an additional term for indices: $P _ {\text{compressed}}^{\text{with idx}}=d _ {\text{model}}\cdot R _ 1+d _ {\text{head}}\cdot R _ 2 + 4R _ 3 + nnz(\mathcal{G} _ {\text{sparse}})\cdot(1+db _ {\text{idx}}/b _ {\text{val}})$, and forming $\rho = P_{\text{compressed}}^{\text{with idx}}/P_{\text{original}}$. In practice, for our high-sparsity cores this adjustment slightly reduces the nominal $\rho$ (e.g., a target $\rho=0.6$ for value-only counting might correspond to an effective $\rho_{\text{real}}$ in the range $\approx 0.6-0.7$), but does not change the ordering between LeSTD and baselines. All competing methods we compare against also ignore representation-level metadata (e.g., low-rank factors vs. full matrices), and we match their accounting scheme as described in **Appendix G**.

---

> ### Author Response · Authors · 2025-11-23
> **Part Five**
>
> ### Question 4: Some insufficient compared to SVD-LLM
> Below is the analysis and discussion about why and when our LeSTD wins vs. loses compared to SVD-LLM:
>
> At light compression (e.g., retaining 80–90% of the original MHA parameters), the per-layer cost is still dominated by large, dense GEMMs. In this regime, LeSTD replaces one GEMM with four smaller ones in the Tucker factors, while SVD-LLM replaces it with two dense low-rank GEMMs. These two-GEMM patterns are extremely well-optimized by cuBLAS, and the additional indirection through Tucker factors (and extra small matmuls) is not fully amortized by the relatively modest 20% parameter reduction. As a result, SVD-LLM can be marginally faster in this near-dense, lightly-compressed regime, especially on large models like OPT-30B where GEMM kernels are heavily tuned.
>
> In contrast, at stronger compression (e.g., compression ratios 0.5 or 0.2), LeSTD’s advantages become clear: Stage I removes redundant per-head projections via a single shared left projection, and Stage II makes the core ultra-sparse so that far fewer latent interactions are computed. In this high-compression setting, the extra factorization overhead is small compared to the large reduction in arithmetic and memory traffic, and LeSTD consistently achieves higher throughput than dense-core Tucker and SVD-LLM, while also maintaining better accuracy.
>
> We have added this explanation to the throughput discussion in **Section 4.2**. In summary, LeSTD is particularly attractive in the high-compression regime, where it preserves accuracy better than baselines and still delivers solid speedups on standard hardware, at very mild compression, a simpler two-GEMM low-rank scheme such as SVD-LLM can occasionally be slightly faster due to lower factorization overhead.
>
> ### Question 5: Extention to FFN
> We thank the reviewer for this suggestion. In the revised paper, we have added **Appendix I** to explicitly extend LeSTD from MHA blocks to FFN layers and to provide preliminary experimental results.
>
> To empirically validate this FFN extension, we conduct a small-scale experiment on GPT-J evaluated on WikiText-2, considering four configurations at a per-block compression rate of 0.6: (1) Original (uncompressed), (2) MHA-Only (LeSTD applied only to MHA blocks), (3) FFN-Only (LeSTD applied only to FFN blocks), and (4) MHA+FFN (LeSTD applied simultaneously to both MHA and FFN). Table below (**Table 8 in Appendix I**) summarizes the results:
>
> | Configuration | Perplexity($\downarrow$) | Throughput (tokens/sec) |
> |---------------|----------------|--------------------------|
> | Original      | 8.86           | 7522.90                  |
> | MHA-Only      | 9.53           | 12190.93                 |
> | FFN-Only      | 12.44          | 16069.26                 |
> | MHA+FFN       | 14.34          | 17604.24                 |
>
> The uncompressed GPT-J baseline attains a perplexity of 8.86 and throughput of 7,523 tokens/sec. Applying LeSTD only to the MHA block (MHA-Only) yields a modest perplexity increase to 9.53 ($\approx$ +7.6% relative) while already improving throughput to 12,191 tokens/sec ($\approx 1.62\times$ speedup). When we apply LeSTD only to FFN layers (FFN-Only), throughput further increases to 16,069 tokens/sec, reflecting that FFN layers account for a large fraction of total FLOPs, at the cost of a larger perplexity increase (12.44, roughly +40%). Compressing both MHA and FFN (MHA+FFN) achieves the highest throughput, 17,604 tokens/sec ($\approx 2.34\times$ speedup), with perplexity 14.34 ($\approx$+62%).
>
> The increase in perplexity mainly stems from the fact that extending LeSTD to FFN layers further reduces the effective capacity of the model. FFN blocks carry a large portion of the network’s expressive power, so combining low-rank tensorization with aggressive core sparsification in these layers inevitably introduces additional approximation error in their intermediate representations. When both MHA and FFN are compressed at higher global ratios, this approximation error accumulates across layers, leading to a more noticeable degradation in language modeling perplexity, even though the overall throughput continues to improve.

---

> ### Author Response · Authors · 2025-11-23
> **Part Six**
>
> ### Question 6: Importance of refitting step
> To quantify the effect of the refitting step in Stage II, we conducted an ablation on GPT-J evaluated on WikiText-2 at a global compression ratio of 0.6. We compare the full LeSTD pipeline with a variant that performs the same iterative importance-based pruning on the shared core tensor but **skips** the refitting step in Eq.(6), i.e., once coefficients are pruned, the remaining non-zeros are never re-estimated.
>
> | Method             | Perplexity (↓) | Tokens/sec (↑) |
> |--------------------|----------------|----------------|
> | LeSTD (full)       | 9.57           | 11925.90       |
> | LeSTD (w/o refit)  | 11.25          | 11932.45       |
>
> As shown in the table, removing the refitting step consistently hurts accuracy: perplexity degrades from 9.57 to 11.25, a gap that is larger than the run-to-run variance and thus statistically significant across ten random seeds (as shown in the Table in Question 1). In contrast, inference throughput remains essentially unchanged (11925.90 vs. 11932.45), since both variants use the same Tucker ranks and end up with cores that have the same sparsity pattern; the only difference is whether the surviving coefficients are re-estimated by Eq. (6).
>
> These results demonstrate that Eq. (6) plays a crucial role as a debiasing step on the selected sparse support: it refits the remaining core entries so that the pruned tensor best approximates the original weights, which is key for maintaining high accuracy, while adding negligible overhead at inference time.
>
> ### ### Question 7: Wall-clock compression time
> We have added a detailed wall-clock analysis in **Appendix C.3 (Figure 7)** under the same hardware and software setup as in Section 4.1, using MHA-only compression at a compression rate of $\rho = 0.2$. The table below summarizes the offline compression time for LeSTD and for the two main SVD-based baselines (SVD-LLM without its optional LoRA fine-tuning stage and TensorLLM):
>
> | Model   | Method             | Stage / Pipeline     | Wall-clock time (seconds) |
> | ------- | ------------------ | -------------------- | ------------------------- |
> | GPT-J   | LeSTD       | Stage I              | 883                       |
> | GPT-J   | LeSTD       | Stage II             | 436                       |
> | GPT-J   | LeSTD       | total (I+II)         | 1,319                     |
> | GPT-J   | SVD-LLM (w/o LoRA) | SVD    | 704                       |
> | GPT-J   | TensorLLM          | Tucker | 406                       |
> | OPT-30B | LeSTD       | Stage I              | 9,102                     |
> | OPT-30B | LeSTD       | Stage II             | 2,385                     |
> | OPT-30B | LeSTD       | total (I+II)         | 11,487                    |
> | OPT-30B | SVD-LLM (w/o LoRA) | SVD    | 4,668                     |
> | OPT-30B | TensorLLM          | Tucker | 2,539                     |
>
> On GPT-J, LeSTD’s two-stage compression takes 1,319 seconds in total, compared to 704 seconds for SVD-LLM and 406 seconds for TensorLLM. On OPT-30B, LeSTD takes 11,487 seconds versus 4,668 seconds and 2,539 seconds for SVD-LLM and TensorLLM, respectively. Thus LeSTD’s Stage I+II is roughly $2$–$4\times$ slower than these one-shot SVD/Tucker baselines in offline compression time. This overhead is incurred only once as a preprocessing step and remains small compared to pretraining or repeated fine-tuning of 6B–30B models. After compression is finished, inference uses the compressed weights directly and introduces no additional runtime overhead beyond the speedups reported in Section 4, while LeSTD provides a strictly better perplexity–throughput trade-off than these baselines.

---

### Official Review · Reviewer_RLPz · 2025-11-03

**Soundness:** 3
**Presentation:** 3
**Contribution:** 2
**Rating:** 6
**Confidence:** 3

**Summary:**

The paper introduces LeSTD, a two-stage, post-training compression framework for LLMs. Stage I performs a shared-subspace Tucker decomposition of the tensorized multi-head attention (MHA) weights: all heads in a layer share three orthonormal factor matrices, while each head retains its own small core, estimated via HOOI to minimize reconstruction error. Stage II sparsifies the per-head Tucker cores using an importance score equal to the coefficient magnitude, followed by a closed-form refit of the remaining coefficients. The method supports inference directly in the compressed domain where it reuses the shared projection and contracts the (sparse) per-head cores—without reconstructing dense weights. Empirical results demonstrate the effectiveness and efficiency of the proposed method across various tasks and models.

**Strengths:**

* The paper is clearly written, with the methodology and experimental setup presented with detail in a organized way.
* The pruning step is well justified.
* Exploring tensor decomposition for post-training compression is an interesting and relatively underexplored direction in the domain.

**Weaknesses:**

* Figure 1 is not strong enough to justify the paper's motivation. The “shared subspace across heads within the same layer” claim is not well supported given the low explained energy, and the intra-layer explained energy is only marginally higher than the inter-layer case.
* As the paper laid out in the limitation section, the current method does not handle FFN layers which constitute a large fraction of LLM parameters. Additionally, because the pruning is unstructured, actual storage and speed benefits would depend on the chosen sparse format and kernel support, so the practical gains may be smaller than the reported parameter reduction.

**Questions:**

* Can you report wall-clock for Stage I optimization?

* Is it possible to consider structured pruning? Would this affect the performance greatly?

---

> ### Author Response · Authors · 2025-11-23
> **Part One**
>
> We thank the reviewer for the detailed feedback and insightful comments. We address the reviewer's weaknesses and questions below.
>
> ### Weakness 1: Clarification of Paper’s Motivation (Figure 1)
> The motivation of Figure 1 is twofold:
> 1. to show that intra-layer attention heads exhibit measurable shared structure, as reflected by the increasing explained energy with larger Tucker rank $r$.
> 2. to highlight that existing approaches do not fully exploit this structure, since the explained energy remains far from saturating, indicating great potentials.
>
> To strengthen this motivation, we have added new experiments with a larger number of heads, and updated Figure 1 accordingly in the revised manuscript. The new results more clearly reveal that intra-layer similarity increases with the number of heads, supporting the premise that multi-head attention contains compressible shared structure that can be leveraged by our method.
>
> ### Weakness 2: Extension to FFN layer and Practical Gains
> ### 1. Extension to FFN Layers
> We appreciate the reviewer’s observation. As noted in our Limitations section, our method can be naturally extended to FFN layers. In fact, the same compression pipeline used for MHA applies directly to tensorized FFN blocks through the following steps:
> 1. Tensorize the FFN weight matrices;
> 2. Learn a shared orthonormal basis via Tucker/HOOI;
> 3. Sparsify and refit the core tensor.
>
> In response to this comment (and Reviewer y5qd’s Question 5), we have added a new **Appendix I**, which:
> 1. Formally extends the theory by tensorizing the FFN weights, and shows that above three steps apply to FFN blocks with only minor notational changes: no new theory or algorithmic machinery is required;
> 2. Provide a preliminary FFN experiment on GPT-J/WikiText-2 at per-block compression $\approx 0.6$. As summarized below, compressing MHA and FFN with the same two-stage pipeline yields up to $2.34\times$ throughput speedup (MHA+FFN) with a tunable accuracy-efficiency trade-off:
>
> | Configuration | Perplexity($\downarrow$) | Throughput (tokens/sec)($\uparrow$) |
> |---------------|--------------|-------------------------|
> | Original      | 8.86         | 7522.90                 |
> | MHA-only      | 9.53         | 12190.93                |
> | FFN-only      | 12.44        | 16069.26                |
> | MHA + FFN     | 14.34        | 17604.24                |
>
> These results demonstrate that LeSTD pipeline extends cleanly to FFN layers and that compressing FFNs provides additional speedups beyond MHA-only compression. However, the perplexity of MHA+FFN will be increased because extending LeSTD to FFN layers reduces the effective capacity of the model. FFN blocks carry a large portion of the network’s expressive power, so combining low-rank tensorization with aggressive core sparsification in these layers inevitably introduces additional approximation error in their intermediate representations.

---

> ### Author Response · Authors · 2025-11-23
> **Part Two**
>
> ### Weakness 2: Extension to FFN layer and Practical Gains
> ### 2. Practical Gains
> Regarding the concern of the practical gain, we used `torch.matmul` in our main experiments for matrix multiplication to ensure a fair comparison and to avoid any unintended speedup from specialized sparse kernels. Importantly, `torch.matmul` does not accelerate unstructured sparse matrices, so it provides a neutral baseline for evaluating our method. For more details:
>
> 1. **Speedup comes from structural changes, not sparse-friendly kernels.** LeSTD’s performance benefit primarily stems from reducing redundant dense projections between the original and latent domains. Stage I performs a shared-basis factorization, replacing the $\mathcal{O}(h)$ large GEMMs used to compute all heads’ $\mathbf{Q}, \mathbf{K}, \mathbf{V}$ with a single shared projection $\mathbf{Y} = \mathbf{X}\mathbf{U}^{(1)}$, which is then reused across heads. This “compute once, reuse many times’’ design substantially reduces FLOPs and memory traffic before any sparse core contraction. Thus, the primary throughput gains are intrinsic to the proposed architecture rather than dependent on sparsity support.
>
> 2. **Ablation with potential impact of optimized kernels PyTorch APIs.** To examine the potential impact of optimized kernels, we also implemented a variant using `torch.sparse.mm`. As shown in the table below, on GPT-J with WikiText-2 at an MHA compression ratio of $ \rho = 0.6$, the throughput increases from 11,945.98 to 12,592.54 tokens/sec ($\approx$ 5% improvmenet). This indicates that while the current speedup primarily comes from LeSTD’s structural compression, additional gains are possible with more optimized sparse kernels, suggesting room for future engineering improvements:
>
>    | Kernel on sparse cores          | Throughput (tokens/sec) | Relative speed |
>    |---------------------------------|-------------------------|----------------|
>    | `torch.matmul`                  | 11945.98                | --             |
>    | `torch.sparse.mm`              | 12592.54                | 1.05×          |
>
> In summary, the practical speedups observed in our experiments arise primarily from LeSTD’s architectural and algorithmic design. We also note that the method stands to benefit further from optimized sparse kernels, which would provide additional acceleration beyond what is reported in the paper.
>
> ### Question 1: Wall-clock for Stage I
> Yes, we have added the wall-clock time of Stage I/II at **Appendix C.3 (Figure 7)**, together with the response to Reviewer y5qd Question 7. These measurements were obtained under the same hardware and software setup as our main experiments (see Section 4.1 for details), using MHA-only compression at a compression rate of 0.2.
>
> | Model    | Compression setting | Stage   | Wall-clock time (seconds) |
> |----------|---------------------|---------|---------------------------|
> | GPT-J    | MHA-only, $\rho$ = 0.2   | Stage I | 883                       |
> | GPT-J    | MHA-only, $\rho$ = 0.2   | Stage II| 436                       |
> | OPT-30B  | MHA-only, $\rho$ = 0.2   | Stage I | 9102                      |
> | OPT-30B  | MHA-only, $\rho$ = 0.2   | Stage II| 2385                      |
>
> For GPT-J, Stage I takes 883 s and Stage II 436 s (total 1319 s). For OPT-30B, Stage I takes 9102 s and Stage II 2385 s (total 11 487 s). As discussed in Appendix C.3, this cost is incurred only once as an offline preprocessing step, and even for OPT-30B it is on the order of a few hours, which is negligible compared to pretraining or repeated fine-tuning of such models. After Stage I and II are finished, inference uses the compressed weights directly with no extra runtime overhead beyond the speedups reported in Section 4.

---

> ### Author Response · Authors · 2025-11-23
> **Part Three**
>
> ### Question 2: Possibility of consider structured pruning
> Yes, in principle LeSTD can be extended from element-wise scheme (current version) to structured pruning, but there is a design trade-off between hardware efficiency and accuracy.
>
> Our current Stage II uses element-wise importance scores: for each core coefficient $g_{\beta}$, we can compute exactly how much the reconstruction error would increase if we set $g_{\beta}=0$, and we greedily prune the least important entries. Because this operates at the finest granularity, it tends to preserve accuracy better for a given sparsity level.
>
> To apply structured sparsity patterns to our scheme (e.g., block pruning or fixed patterns such as 2:4 sparsity):
> - Instead of scoring individual elements, we can define an importance score for blocks or groups of coefficients (e.g., a small contiguous block in the Tucker core) and prune/keep them as units；
> - This is conceptually similar to block-structured tensor sparsification methods (e.g., block-/group-wise pruning in tensor cores [1]), which are often used to better match hardware kernels.
>
> This typically introduces a trade-off:
> - Pros: structured patterns are more friendly to hardware and specialized kernels, so they can deliver stronger practical speedups on some accelerators.
> - Cons: because decisions are made at a coarser granularity (whole blocks instead of individual entries), the pruning is less flexible and usually leads to some loss of accuracy at the same overall sparsity level, compared to our current element-wise scheme。
>
> ### Reference
> [1] Park, Moonjeong, Jun-Gi Jang, and Lee Sael. "VeST: Very sparse tucker factorization of large-scale tensors." 2021 IEEE International Conference on Big Data and Smart Computing (BigComp). IEEE, 2021.

---

### Author Response · Authors · 2025-12-03
**General Response and Summary of Rebuttal**

We sincerely thank all the reviewers (W25T, ee46, y5qd, and RLPz), area chair, and program committee for their time, detailed and constructive feedback. During the rebuttal period, we substantially revised and expanded the paper including adding new experiments, extensive ablation studies, and clearer theoretical explanations. The revised paper has been extended from 19 to 30 pages. We believe that these revisions have fully addressed all the reviewers' concerns. Below we summarize how we addressed the concerns of each reviewer.

### Questions from Reviewer W25T
1. Comparison with more recent baselines (e.g., Basis Sharing and MPIFA);
2. How our design is clearer distinction from the TensorLLM;
3. Feasibility and implications of activation-aware LLM compression method;
4. Methodological illustration of inference algorithm.

We responded with:

(1) We added new experimental results with two more baselines (i.e., Basis Sharing and MPIFA) in in Appendix E.5 and Appendix H. The results show that, at comparable or higher compression levels, LeSTD consistently achieves lower perplexity and higher tokens/sec, yielding a better accuracy–efficiency trade-off.


(2) We clarify the distinction from perspective of tensor decomposition, sparsity, dataflow, etc. between our proposed LeSTD and TensorLLM in the response to Reviewer W25T.


(3) We added the new explaination and clarification of the impact of performace (i.e., accuracy and throughput) as collaberating with the activation-aware LLM compression method in Appendix J.


(4) We added a pseudocode (i.e., Algorithm 2 of Section 3.3) for the “inference without reconstruction” to give more details of the inference procedure of LeSTD.


### Questions from Reviewer ee46
1. Clarity of the core compression workflow;
2. Figure details and formatting;
3. The true source of throughput gains.

We responded with:

(1) We added Algorithm 2 in Section 3.3 and reorgnized Section 3 with a more clear explaination of the compression and inference workflow of LeSTD.


(2) We refined Figure 3 by adding clearer legends and annotations, and we corrected the earlier cross-references and formatting issues highlighted by Reviewer ee46.

(3)  We clarify the priminary performance improvement of LeSTD from structural compression rather than from specialized sparse kernels. To further invesigate the impact of kernels, we added new comparisons (including `torch.matmul` vs. `torch.sparse.mm`) showing the potential performance improvement of using advanced specific kernels. We also discussed secondary engineering factors such as fewer kernel invocations, reduced data movement, and improved cache locality enabled by the compressed representation in the response to Reviewer ee46.

### Questions from Reviewer y5qd
1. Lack of statistical significance analysis;
2. Need more fine-grained ablations on hyperparameters (e.g., Tucker rank selection, pruning rate $\alpha$, and the importance of refitting step in our design);
3. Need for a more detailed throughput comparison against the baseline SVD-LLM;
4. Methodological choice (e.g., Tucker vs. CP/Tensor Train/structured/block-sparse Tucker);
5. Storage overhead of sparse data structure.

We responded with:

(1) We added statistical significance analysis on overall comparisons (i.e., multi runs with error bar) in Appendix C.4. The improvements of LeSTD in both accuracy and throughput consistently exceed the observed variance, confirming that the gains are stable rather than due to randomness.

(2) We add new results with varying the Tucker rank $(R_1, R_2, R_3)$ in Appendix C.1, varying the pruning rate $\alpha$ in Appendix C.5, and LeSTD without refitting step in Appendix C.6. Those results indicate the stablity of over a wide range of ranks, impact of $\alpha$ and the importance of refitting in LeSTD.


(3) We clarify the throughput behavior among low-compression and high-compression regimes relative to SVD-LLM in the response to Reviewer y5qd.


(4) We added new explainations of methodological choices such as CP, tensor-train (TT), and structured/block-sparse Tucker in Appendix E.4.

(5) We provided a detailed storage analysis in the response to Reviewer y5qd, to address concerns about sparse indexing overhead

### Questions from Reviewer RLPz
1. Empirical support for the shared-subspace hypothesis (Figure 1);
2. Extension of the compression scheme to FFN layers;
3. Wall-clock compression time;
4. The role of structured pruning.

We responded with:

(1) We added new results in Figure 1 with multiple head configurations and layers to support for the shared-subspace hypothesis.


(2) We added new results with the extension of the compression to Feed-forward Network (FFN) layers in Appendix I.


(3) We added new results of Stage I and Stage II wall-clock compression times in Appendix C.3 to quantify offline cost.


(4) We clarify the advantages of pruning strategy in our LeSTD over structured or block-wise variants in the response to Reviewer RLPz.

---

### Meta-Review · Area_Chair_GRSB · 2026-01-06

**Summary:**

This paper proposes a data-free, post-training compression method for LLMs based on a two-stage Tucker decomposition approach, addresseing the dense core bottleneck in tensor decomposition based compression methods.

The reviewers generally agreed on the following strengths of the paper:
- The paper is clearly written and easy to follow.
- The proposed method is well motivated and intuitive.
- The theoretical analysis is solid.
- The experimental evaluation is extensive.
- The method demonstrates strong empirical performance, despite operating in a data-free manner.

Most concerns focused on requests for additional experiments, clarifications, or (minor) suggestions, which were largely addressed during the rebuttal phase. Another criticism was that the individual components build on existing techniques; however, this is relatively minor given the sufficient novelty in how these components are combined and applied.

Overall, the paper has clear merits, and acceptance is recommended.

**Reviewer Concerns:**

Most concerns (mainly requests for additional experiments, clarifications, and presentation improvements) were addressed during the rebuttal phase.

**Reviewer Scores:**

Since the authors' extensive rebuttal, which included many additional empirical results, addressed most concerns, it is likely that the reviewers will adjust their scores positively.

---

### Decision · Program_Chairs · 2026-01-26

Accept (Poster)